# Design, Development and Validation of an Intelligent Collision Risk Detection System to Improve Transportation Safety: The Case of the City of Popayán, Colombia

**Santiago Felipe Yepes Chamorro** [1] , **Juan Jose Paredes Rosero** [1] , **Ricardo Salazar-Cabrera** [1,*] ,
**Álvaro Pachón de la Cruz** [2] **and Juan Manuel Madrid Molina** [2]

1   Telematics Engineering Research Group (GIT), Telematics Department, Universidad del Cauca,
    Popayán 190001, Colombia
2   Information Technology and Telecommunications Research Group (I2T), ICT Department, Universidad Icesi,
    Cali 760001, Colombia
*   Correspondence: ricardosalazarc@unicauca.edu.co

**Abstract:** Several approaches from different perspectives have been used to solve problems with traffic accidents (TA), which mainly affect low- and middle-income countries. Conditions of certain cities, regarding road infrastructure, enforcement of traffic safety regulations, and motor vehicle numbers, influence the increase in TAs. Therefore, medium-sized cities in developing countries (context of interest), which commonly have worrying conditions, are a relevant scenario. One of the approaches to reduce TAs has been the use of data analysis through Machine Learning (ML); however, these techniques require a large amount of data, and medium-sized cities commonly do not have enough. Techniques such as Naturalistic Driving (ND) can be applied as a data collection method. This work proposes an intelligent collision risk detection system (ICDRS) using ND and ML to improve sustainability and safety of transportation in medium-sized cities. The ICRDS design considered the limitations of the context of interest and uses two data collection devices in the vehicle. The ICRDS validation included the design and execution of tests using ND. This validation identified if the collected data in a certain time interval contained high-risk collision events (sudden acceleration, sudden braking, aggressive left or right turn, aggressive left or right lane change). The system implementation results were satisfactory. The developed ML algorithm obtained an average value 0.98 in all the metrics. Two data sets of driving on routes were collected. In addition, the performed tests were able to identify city areas with high accident rates.

**Keywords:** collision risk detection; machine learning; naturalistic driving; near-crash; sustainability and safety in transportation; traffic accidents

## 1. Introduction

According to the Global status report on road safety presented by WHO, the number of worldwide annual road traffic deaths reached 1.35 million in 2016. Some of the reasons for this death toll are rapid urbanization, poor safety standards, lack of enforcement, driver distraction or fatigue, driving under the influence of drugs or alcohol, speeding and a failure to wear seat belts or helmets [1]. Road traffic injury is the leading cause of death for people between 5 and 29 years old, and it is the eighth leading cause of death for all age groups [1]. More than a half of global road traffic deaths involve pedestrians, cyclists and motorcyclists, who are still often neglected in road traffic system design in many countries. This issue is mainly increasing in developing countries [1,2].

The World Bank categorizes world's economies into four income groups—low, lower-middle, upper-middle and high-income countries. Low and middle-income countries include the first three of the four groups. Every year, the World Bank classifies the countries based on their GNI (Gross National Income) per capita in US Dollars [3].

A developing country is defined as a country that has an annual per capita gross national product (GNP) less than 9361 American dollars, according to World Bank. Most low and middle-income countries fall into this category [2].

The reasons for high burden of road traffic injuries in developing countries are the growth in motor vehicle numbers, the number of people killed or injured per crash, poor enforcement of traffic safety regulations, inadequacy of public health infrastructure, and poor access to health services [2]. Another relevant factor is road infrastructure, because developing countries usually have a large number of motor vehicles, but their road conditions are not the best [4].

Crashes are the most common type of traffic accidents (TA), and one of the main causes of road traffic injuries [4]. Therefore, to improve sustainability and safety in transportation of cities in developing countries (from now on, the context of interest), it is essential to take actions aiming to minimize the occurrence of these events. By seeking the prevention of TAs, the negative impact on the social (deaths and injuries) and economic (cost of a TA) aspects of the sustainability in the context of interest could be reduced.

Conditions of certain cities in developing countries (e.g., Colombia) concerning road infrastructure, compliance with traffic regulations, and vehicle operation condition, further affect the occurrence of TA [5,6]. Medium-sized cities in Colombia have worrying conditions in these aspects. The American Community Survey (ACS) divides cities into four groups according to their size—small cities (between 50,000 and 99,999 residents), small-mid-sized cities (100,000 to 149,999 residents), middle-sized cities (150,000 to 499,999 residents), and large cities (500,000 residents or more) [7].

The number of available roads in Colombian medium-sized cities is commonly low, and such roads do not have adequate maintenance. In addition, public transit vehicles share the road with private vehicles. Speed limits and red lights are commonly violated, mainly because technological enforcement of regulations is relatively low. Finally, a large percentage of the vehicles is rather old and has no adequate maintenance [5,6].

Road traffic injuries are a global major cause of death and disability, with a disproportionate number occurring in developing countries [2]. The injury profile for road traffic crashes in developing countries differs in important ways from the profile seen in developed countries, with annual mortality rates of 24.1 per 100,000 inhabitants in developing countries, compared to 9.2 in developed countries [4].

This work is focused on medium-sized cities in developing countries, because the scenario of these countries has worrying characteristics in terms of mobility and high rate of TAs. Moreover, medium-sized cities of developing countries have more TA-related worrying issues than other cities in these countries. In Colombia, large-sized cities have TA death rates close to or below the national rate (14.28 in 2015 per 100,000 inhabitants), meanwhile medium-sized cities significantly exceed national rate. One example of this is Popayán, which had a rate of 21.26 deaths per 100,000 inhabitants in 2015 [8].

Another reason to focus this work on developing countries is found in the current research gap related to road safety. According to Haghani et al. [9], only 10% of the investigations related to road safety took into account the context of developing countries.

The context of the proposed work generates some limitations regarding the proposed development due to limitations in resources, available budget, technology for data collection and number of performed tests; however, the results can be applied to large-cities in developing countries, or to any city in any country.

Data analysis has gained strength in recent years to identify risk factors or propose solutions to problems related to TAs [10,11], using different technologies such as Internet of Things (IoT), data mining and Machine Learning (ML). In recent years, ML has been widely used as a tool for road safety applications, traffic control and autonomous vehicles [12–14]. This approach could be used in the context of interest, albeit in many cases there is not enough available accident data to extract relevant information, or such data may not exist. Methods such as Naturalistic Driving (ND) allow collecting data to analyze collision risk events in scenarios where there is not enough collected accident data. Thus, ND can be

applied in the context of interest, considering that the method's complexity and the amount of required resources are manageable [15].

ND provides information on TAs [16] commonly using a vehicle's kinematic data to detect high-risk driving behaviors that could cause an accident [16,17]. This work proposes the collection of driving data through ND, focusing on identifying certain driving events or maneuvers that could generate an accident (sudden acceleration, sudden braking, aggressive left or right turn, aggressive left or right lane change), as suggested by works like [13,18,19], instead of identifying certain types of collisions.

Furthermore, some ND approaches complement kinematic data with continuous video to monitor driver behavior and performance [17]. In this work, it is important to consider that the two significant variables regarding driving behavior are the vehicle's kinetic energy and average deceleration [16]. These two variables can be measured by sensors inside the vehicle, without the need for continuous video.

In ND studies, the concept of near-crash is important as a substitute for crash detection in TAs. The risk factors for TAs detecting a near-crash are more sensitive than in crashes [17]. In ND experiments, each participant driver is informed to maintain his/her regular driving style as much as possible. In addition, to ensure that the experiments were standardized and conducted in similar conditions, the participants were asked to drive on the same routes using the same data collection device in the vehicle, during the same periods [16,17]. The number of participants, routes and trips in the ND experiments is limited, depending on the context where it is applied.

ND experiments collect a wide variety of driving variables and/or driver behavior, using several types of data collection devices [16,17,20–25]. The goal of the ND experiment in this work is to identify if the collected data in a certain time interval contains high-risk collision events. To achieve this goal, it was necessary to determine the driving events in which aggressive or unusual maneuvers related to near-crash events occur. Detailed information about the identified driving events is presented in Section 2.5.1.

In the literature review, some research works performed an identification and study of characteristics that influence crashes using ND data sets [16,17,20–25]. These works used an ND data set to determine, detect, or prevent accidents using ML algorithms or statistics. Most of these works used the ND dataset created by the Second Strategic Highway Research Program (SHRP2) [20–24]. Another proposal used ND data created at Virginia Tech [25]. The other two works used other data sets [16,17].

Other revised works in the literature present different methodologies, techniques and models to efficiently detect driving risks efficiently [16–18,26–29]. Most of these works conduct a Naturalistic Driving Study (NDS) [16–18,26–29]. Another work does not use any dataset [30], as this one comparatively evaluates ND data collection approaches such as On Board Diagnostics (OBD-II), In-Vehicle Data Recorders (IVDR), smartphone sensor data, among others. Two of these works used the near-crash technique to increase the quality of the results obtained by the NDS [16,17]. All these works used data captured in urban routes and performed tests to validate the concerned methodologies, models and/or techniques.

Some other identified works performed a design and implementation of naturalistic or normal data collection systems and techniques [19,31–34]. The data collection devices included Smartphones, OBD-II, and the vehicle's internal sensors. The most used ML algorithms to analyze the collected data were Support Vector Machines (SVM) and Artificial Neural Networks (ANN). These articles use some type of vehicle data collection technique for different purposes. Three works use Smartphone sensors to collect data [19,31,32]. The other two works use sensors placed inside the vehicle [33,34].

Finally, some other identified works present proposals of ML models or techniques to predict near-crashes or to detect high accident risk areas [35–42]. The most used ML algorithms in these works were SVM and Neural Networks (NN). Some works used data collected on their own, and others used existing data sets. Most of the reviewed works used 70% of the data set for training the ML model and 30% for testing. Four of these works used one or more databases to detect near-crashes. Osman's work used the SHRP2

database [35]; Amiri's used an accident database from Mashhad (Iran) [36]; Tao's used multiple TA databases on major roads in Beijing [37]; and Bao's used New York City's crash data and taxicab route databases [39]. The other four works focus on building their own database to perform ML analysis [38,40–42].

This work proposes an intelligent collision risk detection system (ICRDS) to improve transportation safety in the context of interest. The construction of the system considered the following four phases: (a) Identification of devices, kinematic variables and algorithms for characterizing near-crash events; (b) System architecture design; (c) Design and development of a system prototype, including a data collection module, an intelligent analysis module and an ML model; (d) Design and development of field tests in the target context, for validation of system and algorithm performance. Finally, the obtained results were analyzed, areas with a higher risk of occurrence of a TA in the city taken as a case study were identified, and some future works were proposed.

The principal technical contributions in this article are:

- The design of the ICRDS was performed considering the limitations of the context of interest. The research works reviewed were focused on other types of cities.
- The ICRDS data collection was performed by simultaneously using two types of devices, allowing data cross-validation.
- Test design and execution, and the analysis of results in the context of interest were included.

Additionally, this work produced two important results, in terms of transportation safety:

- Two data sets were collected with relevant information regarding vehicular routes, taken from two devices. High-risk near-crash events were identified in such data sets [43].
- A classification of critical areas with a higher risk of occurrence of TAs in the city used as a case study (Section 3.3).

Considering all the above, the research questions of this work are the following:

- What are the main components of a system that allows collision risk detection in a medium-sized city of a developing country?
- How to collect relevant data on vehicular trips, through an experiment that uses ND, to identify high-risk collision events in cities of the target context?
- How to determine the areas of high probability of TA in the target context?

The rest of this article is organized as follows: Section 2 presents the materials and methods used in the research. Section 3 presents the detailed results of the work. Section 4 discusses the results. Finally, Section 5 presents conclusions and future work.

## 2. Materials and Methods

This section presents the methods used in the development of the research work described in this document. As a summary, Figure 1 describes the methodologies, frameworks, and the general structure of the document to answer each of the research questions.

### 2.1. Identification of Devices, Kinematic Variables and Algorithms

In this phase, initially, a literature review was conducted following the established guidelines in the Preferred Reporting Items for Systematic Reviews and Meta-Analyses (PRISMA) methodology [44]. PRISMA was deemed as a good option, due to the order it establishes and the standardization of its phases. PRISMA has four phases: identification, screening, eligibility, and inclusion.

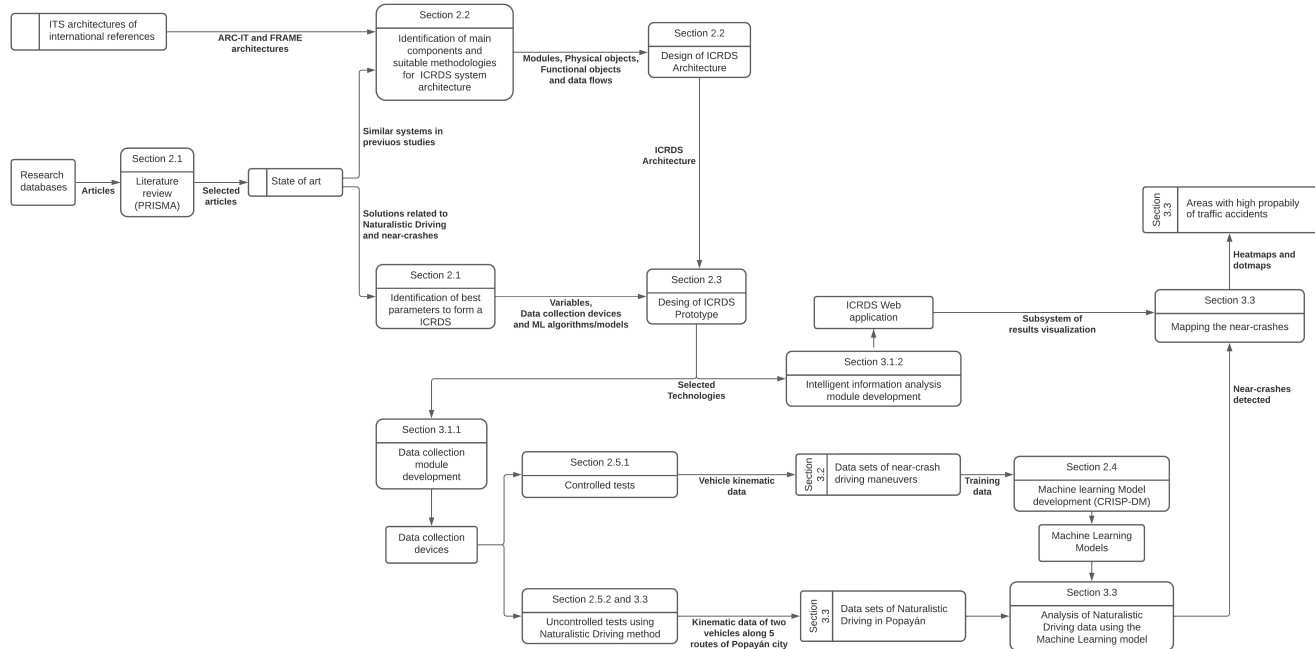

**Figure 1.** Overview data flow chart of the methodology and work structure.

The four phases recommended by the methodology were performed, reducing the number of articles reviewed in each phase. In the identification phase, articles were filtered by year of publication and type. In the screening phase, repeated articles and articles whose title and abstract were not related to presented topics were eliminated. Some criteria related to ND, ML, and accident prediction or prevention were established to select the items that passed the eligibility phase. Finally, in the inclusion phase, the 27 articles were classified into four different groups based on their similarities. A qualitative synthesis was performed for each of these four groups, and a quantitative review was conducted for the last two groups, to identify comparison parameters between the articles.

Each one of the four groups of documents corresponds to a relevant approach for the design of an ICRDS. The four approaches were: most influential variables for near-crash detection; systems or techniques used in data collection; the methodologies, techniques and models useful in crash detection; and the algorithms used for crash prediction or detection.

11 articles were selected from these four groups [16–19,23,30,31,35,38–40]. The selected articles considered the following criteria:

- Use of components adjusted to the context of interest.
- More than a driving variable was considered.
- Field tests in real driving environments were performed.
- Data acquired during the field tests.
- Analysis of the collected data.

The selected articles provided an overview of the variables, devices, and techniques used in solutions related to near-crashes and ND. In addition, it was possible to identify two important modules: a data collection module and an intelligent information analysis module. Three components of interest were selected to form an ICRDS: Variables to be measured, data collection devices, and ML algorithms and models.

Before analyzing the possible variables to be measured, those commonly used in similar systems and those that are more related to the identification of near-crashes with the help of ND were identified in the literature review. Variables selected for the ICRDS were acceleration, speed, yaw rate, magnetic field strength, time stamp, latitude, longitude, and altitude. They can be captured by using an accelerometer, a GPS, a gyroscope, a magnetometer, and an internal clock.

The devices considered to perform the data collection inside the vehicles were:

- Instrumented vehicle.
- Smartphone.
- On Board Diagnostic II (OBD-II) system.
- Inertial Measurement Unit (IMU).

These types of devices were identified in the literature review. For each device, advantages and disadvantages related to ICRDS implementation were analyzed. The Smartphone was selected as the principal alternative for the system. However, considering its disadvantages, it was considered an alternative source of data collection. The Smartphone is the most commonly used device in recent ND studies [30]. This device collects large amounts of data in a short time, the cost of developing a solution with these devices is relatively low, and its use is sustainable and scalable [31]. Its main disadvantage is the amount of noise captured by the sensors [31]; however, this issue could be resolved using filters or cleaning methods [19,31].

For an alternative source of data collection, the instrumented vehicle option was discarded due to the high cost and complexity. The options OBD-II and IMU do not allow capturing all the variables selected for this work, for this reason, the use and development of a hybrid device was considered. This hybrid device uses a microcontroller card for processing the data, and the in-vehicle OBD-II port, an IMU, and a GPS module for data collection. The most suitable option for the microcontroller card was the Raspberry Pi 3, considering some of its advantages: Linux-based operating system; its processor power and amount of RAM memory, which allow different high-processing tasks to be performed; incorporates Bluetooth and Wi-Fi communication, and a general-purpose input and output (GPIO) port for connection of different types of sensors.

ML algorithms selected for the ICRDS were determined by considering three quantitative aspects and a qualitative one. The quantitative aspects were performance, amount of used data and number of implementations. The qualitative aspect was the relationship between the input features, the target, and the goals of this study. The following algorithms were considered: AdaBoost, Decision Tree (DT), Gaussian Naive Bayes (GNB), Random Forest (RF), STCL-Net, and Support Vector Machines (SVM).

After performing the analysis, the selected algorithms for the development of the intelligent analysis module were SVM, RF, and DT. These three algorithms showed good performance in several metrics of a considerable number of reviewed works, they were used in works with a high amount of analyzed data, and also in those works that are highly related to the objectives of this research.

### 2.2. Architecture Design for the System

For the development of the system architecture, it was necessary to perform a review of similar system architectures proposed in other works. This review was performed considering three sources: Architectures found in the resulting articles at the end of the systematic review; architectures from other relevant sources, and Intelligent Transportation Systems (ITS) architectures of international reference.

In the first two groups, most of the articles analyzed were not related to the main objective of this work, but their architectures had characteristics that can be used in the development of the architecture for the proposed system. In the last group, two ITS related architectures were analyzed: Architecture Reference for Cooperative and Intelligent Transportation, ARC-IT [45]; and European Intelligent Transport Systems Framework Architecture, FRAME [46]. Each ITS architecture provided a framework and service packages that helped to build the proposed system architecture.

The objective of this review was to identify the main components of the architecture required for the system and the suitable methodologies for its design. The proposed architecture was the adaptation of several existing architectures, including their most relevant components, and considering the proposed context. Two main modules were

identified for the system: a data collection module and an intelligent information analysis module. Figure 2 presents a first approximation to the ICRDS architecture.

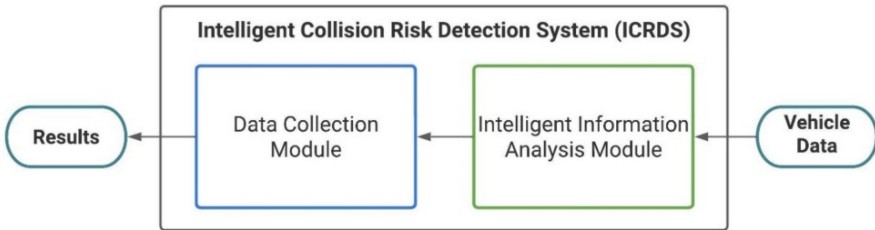

**Figure 2.** First approximation of the ICRDS architecture.

Then, the communication between the two segments was analyzed. To have an online information exchange, a stable and low latency communication would be required. This type of communication could be difficult to achieve in the context of interest because the operation of mobile networks does not have the necessary coverage and quality. Additionally, in the case of using existing mobile networks, the usage cost would also be a factor to consider [47].

Near-crash detection requires measured data to be sent with a high frequency; the loss of data due to inadequate communication would cause a poor performance of the model, and therefore poor results. Considering this, the use of a temporary storage component could be appropriate, because it generates certain advantages such as a reduction in system costs and complexity. Thus, the data analysis module should be implemented as an offline module.

Considering the above, the identification of the proposed collision risks is performed offline. The information stored in the data collection is later transferred to a central cloud storage. Data analysis is performed in the central cloud storage.

The ARC-IT architecture was selected as the model architecture for adjusting and proposing an option for the ICRDS, due to its organization, clarity, standardization and permanent updating. Based on ARC-IT, the segments or modules, physical objects, functional objects and information flows used in the proposed architecture were determined. Some of them were taken directly from the service packages established in ARC-IT. Others were defined or adjusted according to the characteristics of the target context or the necessary system functionalities. The proposed ICRDS architecture is presented in Figure 3.

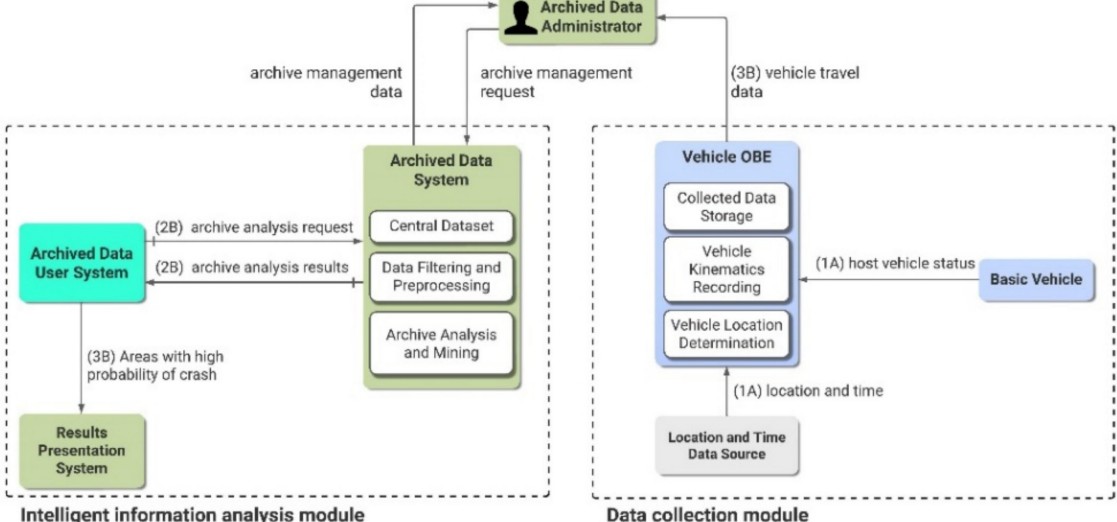

**Figure 3.** Proposed system architecture.

The details of the architecture are presented in Appendix A of this document.

### 2.3. Design and Development of a System Prototype

From the previously proposed architecture, it was possible to design a prototype for the ICRDS, which is presented in Figure 4.

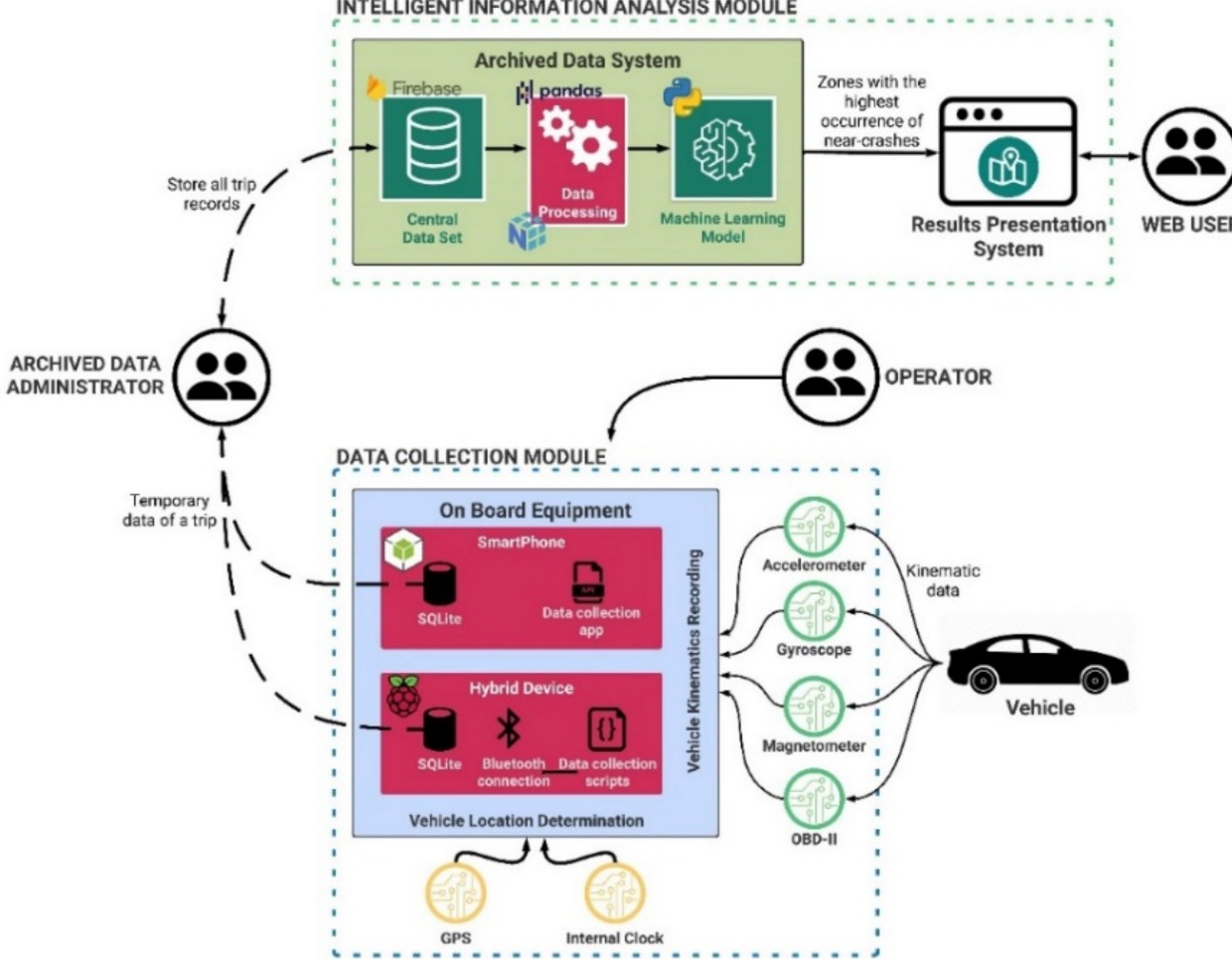

**Figure 4.** Proposed prototype for the ICRDS.

The proposed prototype in Figure 4 features two modules: data collection and intelligent information analysis.

- Data Collection Module (DCM). This module has the necessary hardware (HW) and software (SW) for collection and storage of vehicle kinematic data. This process encompasses three main operations: kinematic variable capture, location and time data tagging, and temporary storage of collected data.
- Intelligent information analysis module (IIAM). This module contains the necessary components for information processing and data cleaning, ML algorithms and presentation of results.

Details of each module of the ICRDS prototype are presented in Appendix B, including their physical objects.

Six functional objects were implemented in the prototype, distributed in the physical objects of each module. These functional objects are:

- OBE—Collected Data Storage (OBE-CDS).
- OBE—Vehicle Kinematics Recording (OBE-VKR).
- OBE—Vehicle Location Determination (OBE-VLD).

- ADS Central Data set (ADS-CD).
- ADS Data Filtering and Preprocessing (ADS-DFP).
- ADS Archived Analysis and Mining (ADS-AAM).
- Details of functional objects of the ICRDS prototype are presented in Appendix C.

*2.4. ML Model Development*

The ML model development for the prototype was based on the use of the CRISP-DM methodology. This methodology has 6 phases that defined the development of the ML model used by the ADS-DFP and ADS-AAM functional objects of the IIAM module.

2.4.1. Phase I: Business Understanding

In this phase, the objectives to be achieved with the development of the ML model, the expected results, and the plan that allowed the achievement of the objectives were determined.

The objective to be achieved with the development of the ML model was to "identify, through vehicle kinematics, areas where driving events related to near-crashes may occur". To achieve the objective, it was necessary to determine driving events in which aggressive or unusual maneuvers related to near-crash events occur.

The plan to meet the goal of ML model development included the following steps:

- Extract data where the near-crash driving maneuvers occur.
- Clean and filter the data obtained when executing the driving maneuvers.
- Select the features from the data set that best represent each maneuver.
- Perform the feature extraction process to determine the final input data set of the classifier algorithm.
- Train and optimize each one of the classifiers to select the one that obtains the best performance classifying the driving maneuvers.
- Select the classifier that obtained the best performance to generate the ML model that allows the detection of the different maneuvers.
- Deploy the classifier on the ICRDS prototype web platform.

2.4.2. Phase II: Data Understanding

In this phase, the activities for familiarization with the data were performed. Data for creation of the ML model was acquired through the execution of controlled tests of abnormal driving maneuvers; details of these tests are presented in Section 2.5.1. The two instances of the data collection module (the Smartphone device and the hybrid device) collected the information with a frequency of 20 Hz. At the end of the tests, a total of 36,353 rows of raw data were collected; each row featured 18 columns for data captured with the Smartphone and 19 columns for data captured with the hybrid device.

The 18 columns for files registered by the Smartphone are as follows:

- Data ID (input);
- Trip ID (input);
- Vehicle ID (input);
- Route ID (input);
- Timestamp (input);
- Speed (input);
- Acceleration in X (input);
- Acceleration in Y (input);
- Acceleration in Z (input);
- Angular velocity in X (input);
- Angular velocity in Y (input);
- Angular velocity in Z (input);
- Magnetometer in X (input);
- Magnetometer in Y (input);
- Magnetometer in Z (input);
- Latitude (Input);

- Longitude (Input);
- Event Class (output). It is a binary variable, where 1 indicates the occurrence of a near-crash and 0 indicates normal driving. All columns are numeric data except for the timestamp and the event class (binary).

The 19 columns for files registered by the hybrid device are the same 18 of the Smartphone, adding the position of the accelerator pedal (0 to 100), which was captured with the help of OBD-II.

To capture data from the Smartphone, its internal sensors and a custom-made application were used. For the hybrid device, a Raspberry Pi 3 computer, an IMU, an OBD-II interface, and a GPS module were used, along with a custom-made application. These components and a developed application (similar to the one made for the Smartphone) were in charge of collecting the features of the hybrid device.

The ML model for the two data collection devices is a classifier which determines if a certain record corresponds to a near-crash or a normal driving state. In the controlled tests, a training and testing process was performed by labeling the events using the Event Class field. In the uncontrolled tests, the previously trained model was evaluated.

The tool used for reading and visualizing the data was the Travel Data Management and Analysis subsystem presented in Section 3.1.2.

Figure 5 illustrates the relationship between each driving maneuver to be detected and the kinematic characteristics stored in the data sets. The red color in the graphs represents the moment in which an aggressive maneuver was executed. As mentioned in Section 2.5.1, test execution was performed in time intervals between 14 and 30 s, while aggressive maneuvers took approximately 0.7 to 2.5 s to complete. The graphic analysis of each driving maneuver allows to hypothesize that each of them could be defined by a subset of the kinematic measured characteristics, such that those characteristics not provided information could be discarded.

In this way, it was possible to demonstrate that the acceleration and sudden braking maneuver presents significant variations in speed and acceleration in the "Y" axis (exceeding acceleration thresholds of 5 m/s$^2$ and $-5$ m/s$^2$, respectively), while the other variables remain indifferent to this type of events, presenting insignificant variations or only noise. For the aggressive turning maneuver, both to the right and to the left, the most relevant characteristics were the acceleration in the "X" and "Y" axes (exceeding acceleration thresholds of $\pm4$ m/s$^2$ and $-5$ m/s$^2$ respectively), the angular speed in the "Z" axis (with thresholds equal to or greater than $\pm0.5$ rad/s) and the magnetic force in the "X" and "Y" axes (presenting rapid changes of $\pm20$ uT). Finally, the aggressive lane change maneuver, both to the right and to the left, presents significant variations in the acceleration of the "X" axis (exceeding acceleration thresholds of $\pm5$ m/s$^2$) and the angular velocity of the "Z" axis (with thresholds equal to or greater than $\pm0.5$ rad/s).

In addition, characteristics that are independent of any maneuver performed on the vehicle were also identified. These are the acceleration in "Z" axis, which is always 1 g, and the angular velocity in the "X" and "Y" axes, which always marked the roll and pitch rotations of the vehicle. The angular velocity of these rotations is not normally strongly affected in a four-wheel drive vehicle.

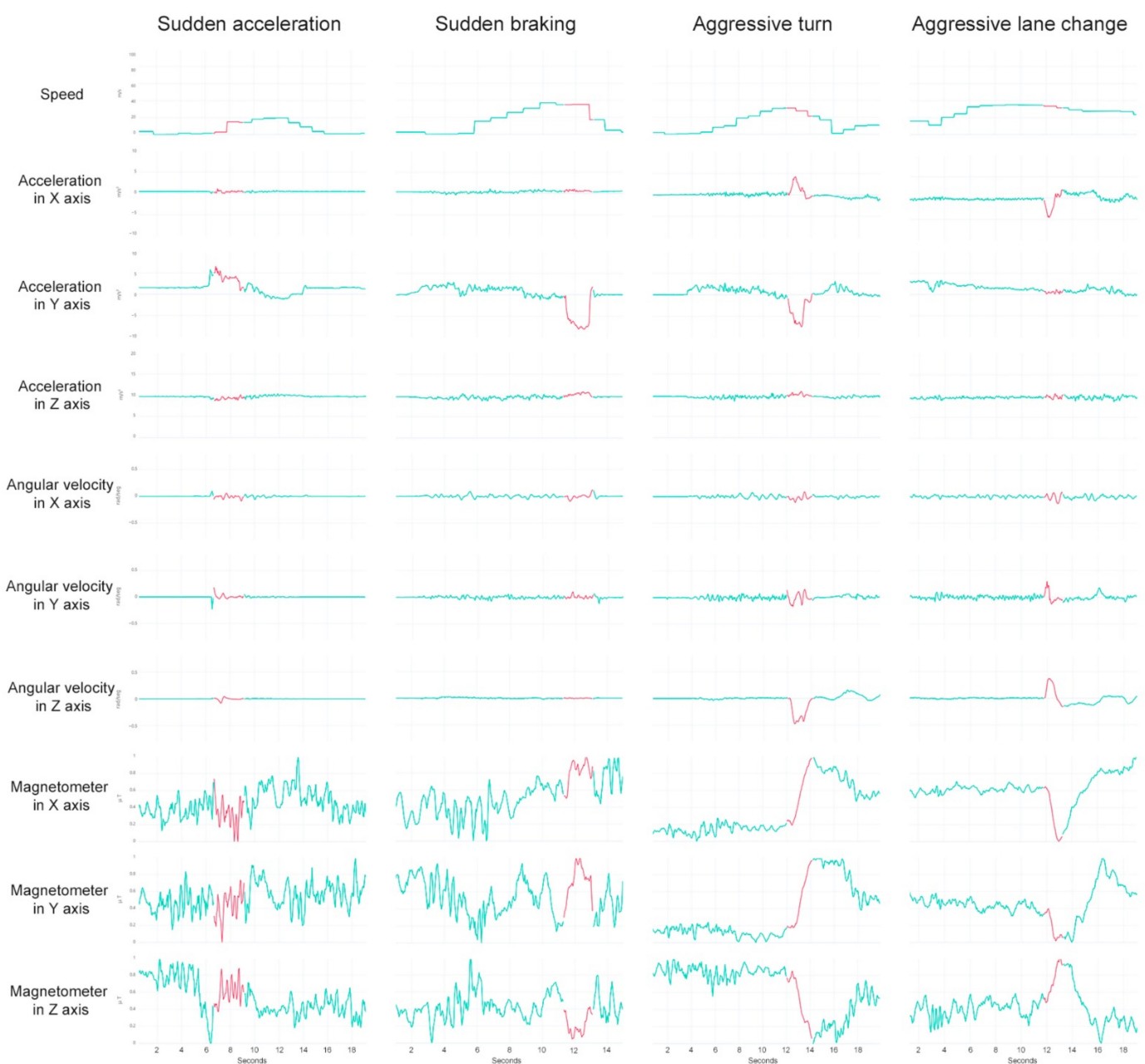

**Figure 5.** Visualization of the relation between kinematic variables and driving maneuvers.

### 2.4.3. Phase III: Data Preparation

In this phase, the activities necessary to generate the final data set served as input to the ML model. The activities of this phase were the following:

- Feature selection. It was necessary to identify the variables that had better correlation in the dispersion matrix, notable changes in the data according to the performed maneuver (Figure 5), and relevant information based on the current state of knowledge. The characteristics totally discarded from the data set with raw data were the acceleration in "Z" axis, because it is constant and equal to 1 g; angular velocity in the "X" and "Y" axes, because the variations observed in the files of the data set were tiny compared to the angular velocity in "Z" axis; and finally, the measurement of the magnetometer in the "Z" axis was discarded, because the "Z" axis always remained orthogonal to the ground plane. With the review performed, it was possible to define the different driving maneuvers from different kinematic variables (Table 1).

- Data cleaning. This activity consisted of two parts, a manual part and an automated part. In the first part, the necessary corrections were made for 3 problems related to data quality. The problems were related to the disconnection of the ELM 327 scanner, the operator mislabeling of a driving event, and a shift in the acceleration data due to the presence of tilt where the devices were placed. The second part of the data cleaning was focused on solving a problem related to the presence of noise in the IMU's data due to factors such as engine or wheel friction. To solve this problem, a wide variety of denoising methods can be applied [14], in the case of this work a Kalman filter was selected. This filter was applied to all the columns that were related to kinematic data using a Python script [18].

- Feature extraction. The sliding time window was the technique used for feature extraction. This type of technique is frequently used to characterize time series data. The sliding time window contains two adjustment features: the size of the window, and the sliding interval. For the development of this work, only the size of the window was varied, taking values of 20 or 40 data samples.

**Table 1.** Summary of feature selection to detect each maneuver.

| Driving Maneuver | Kinematic Features | | | | | | | | | | | |
|---|---|---|---|---|---|---|---|---|---|---|---|---|
| | spd [1] | spd OBD-II [2] | acc Pos [3] | acc [4] X | acc [4] Y | acc [4] Z | vel Ang [5] X | vel Ang [5] Y | vel Ang [5] Z | mag [6] X | mag [6] Y | mag [6] Z |
| Sudden acceleration (Smartphone) | √ | | | | √ | | | | | | | |
| Sudden braking (Smartphone) | √ | | | | √ | | | | | | | |
| Aggressive turn (Smartphone) | | | | √ | √ | | | | √ | √ | √ | |
| Aggressive lane change (Smartphone) | | | | √ | | | | | √ | | | |
| Sudden acceleration (Hybrid device) | | √ | √ | | √ | | | | | | | |
| Sudden braking (Hybrid device) | | √ | | | √ | | | | | | | |
| Aggressive turn (Hybrid device) | | √ | | √ | √ | | | | √ | √ | √ | |
| Aggressive lane change (Hybrid device) | | | | √ | | | | | √ | | | |

[1] spd: speed; [2] spd OBD-II: speed obtained from OBD-II (only for hybrid device); [3] acc Pos: position of the accelerator pedal; [4] acc: acceleration in X, Y or Z axis; [5] vel Ang: angular velocity in X, Y or Z axis; [6] mag: magnetometer in X, Y or Z axis.

Using the sliding time window allows the transformation of the sequential data into sets characterized by a significant measure, this measure consists of mathematical calculations allowing to generate a greater gain of information of the atypical events presented in the driving data associated to a maneuver. The following six transformation measures were defined: mean, median, standard deviation, maximum, minimum and trend.

After characterization, a matrix of measurements is generated for each of the driving maneuvers; this matrix is used as input for the ML model. The dimensions of this matrix are obtained in this way:

$$\text{rows} = \#\text{ of records in the driving maneuver} - \text{window size} + 1 \tag{1}$$

$$\text{columns} = 6 * \#\text{ of kinematic features on the driving maneuver} \tag{2}$$

e.g., for a sudden acceleration maneuver with 382 records and a time window of 40 samples, the measurement matrix will have 343 rows and 12 columns.

### 2.4.4. Phase IV: Modeling

In this phase, the activities related to the construction of the selected classifier algorithms (SVM, RF, and DT, determined in Section 2.1), the optimization of the parameters of each classifier and the creation of classifier performance tests were performed.

To create each of the classifiers, Python's predictive data analysis package (ScikitLearn) was used. Classifiers in ScikitLearn are defined as a class of a specific ML algorithm.

ScikitLearn has classes for the three selected algorithms, which facilitated the required development. The classifiers are defined as classes that have methods such as fit and prediction, through these methods the classifier is enabled to make predictions. 12 classifiers were created in total, 6 for each type of maneuver multiplied by 2 types of devices (smartphone and Hybrid).

From the predictions it was possible to test the preliminary performance of each of the classifiers. In these performance tests, the following evaluation metrics were used: precision, recall, F1-score and Area Under Curve (AUC). The initial performances of the classifiers were not optimal; hence it was necessary to adjust the hyper-parameters of each algorithm. In addition, a cross-validation scheme was used to evaluate the performance of the total data set.

### 2.4.5. Phase V: Evaluation

In this phase, the classifiers generated in the previous phase were evaluated to select the classifier with the best performance. In addition, field tests were used to check if the model met the objective of phase I. Finally, the possible actions to be performed in future iterations were determined.

Evaluation sets were used in the evaluation of each of the classifier together with their hyper-parameters. Each evaluation set was formed as follows: (a) ML classifier, (b) device, (c) driving maneuver, (d) sliding window size and (e) optimized metric.

The design of the controlled tests to collect the data set to train and to evaluate the classifiers is presented in Section 2.5.1, while the results of such tests are presented in Section 3.2. These results allowed to define the classifiers that had the best performance when classifying the different driving maneuvers, as well as the time window and the hyper-parameters yielding better performance.

In this way, the RF algorithm and a time window of 40 samples were chosen for the development of the classifiers in the ML model.

### 2.4.6. Phase VI: Deployment

In the last phase of CRISP-DM, the way in which the ICRDS web application would use the ML model was developed. For this, it was necessary to use the joblib tool that allowed to create persistent ML classifiers on disk [48].

With joblib, the 12 ML classifiers (a classifier for each of the 6 types of driving maneuvers, and for each of the two devices) were generated in files so that the ICRDS web platform would later load and execute them. The detection is done in this way: (a) the data is downloaded from the central database; (b) the data preparation process is performed (Kalman filter); (c) the 12 classifiers determine the occurrence of a maneuver; (d) each detection of a maneuver is stored as a near-crash; (e) repeated near-crash events are joined; and f) the events are sent to the central database.

In addition, uncontrolled tests were performed for the validation of the ICRDS prototype. The design of the controlled tests to collect the data set to evaluate the classifiers is presented in Section 2.5.2, while the results of such tests are presented in Section 3.3.

### 2.5. Design and Development of Field Tests in the Target Context

The developed field tests were designed based on the ML model development process. These tests were required to validate the ICRDS operation in the identification of near-crashes in the routes and, consequently, the areas of high accident rate in the city used as a case study. This led to designing and conducting two types of field tests: controlled and uncontrolled.

The controlled tests were focused on the training and creation phase of the ML model. The uncontrolled tests were aimed at validating the operation of the previously trained ML model, using the ND methodology and the detection of near-crashes using the developed ML model. The controlled tests do not represent a realistic scenario, since the maneuvers that were intended to be identified were performed to train the ML algorithm. However,

the uncontrolled tests represent a realistic scenario, since they used the ND method. In this realistic scenario, the operation of the developed ICRDS was validated. Data was first collected through the planned routes, and then analyzed with the previously developed and trained ML model.

### 2.5.1. Controlled Training Field Tests

The implemented ML model aimed to solve a classification problem through supervised learning. The classification made with the ML model should allow identifying if the data collected in a certain period of time was a near-crash event or not. To achieve this goal, it was necessary to determine the driving events in which aggressive or unusual maneuvers related to near-crash events occur. The selected maneuvers were sudden acceleration, sudden braking, aggressive left or right turn, aggressive left or right lane change. These maneuvers were the most used in related works, and best characterized a high-risk driving maneuver [17–19,23,25,39].

Based on the above, the controlled tests were designed, which were aimed at collecting driving data that included the maneuvers of interest and normal driving events. The desired result with these tests was a training data set with labeled data, a required element in all supervised learning problems. This data set would be the principal source for the development of the ML model.

For the execution of these tests, a Nissan March model 2016 vehicle and a single driver were used, repeatedly driving short distances (driving times between 14 to 30 s). The vehicle used for the tests was selected among the resources available for the project considering its compatibility with the OBD-II module. The maneuvers were performed imitating the kinematics each of them would have during a driving risk situation, as realistically as possible. These tests were performed on flat roads with almost no vehicle traffic. This helped to avoid possible accidents during the execution of the different maneuvers due to their high risk. The smartphone and hybrid devices were in the middle area of the passenger floor, inside the vehicle.

The frequency selected for the execution of the tests was 20 Hz, the reason for this selection was based on the time it took to perform each of the maneuvers, approximately 0.7 s to 2.5 s, so obtaining 20 samples per second would be sufficient to meet the Nyquist sampling rate condition [24]. In addition, preliminary tests showed that a high frequency for data capture (50 Hz or more) produced instability errors both in the application and in the IMU device sensors.

The label in the collected data represents the occurrence or not of a maneuver of interest and therefore of a near-crash. To allow labeling, both collection devices were developed with a functionality that would facilitate this task. In the case of the Smartphone, this was possible through a button located in the main interface of the application; in the case of the hybrid device, by means of a pushbutton connected to a GPIO pin of the Raspberry board.

During the data collection process, the described buttons were pressed by the DCM operators when the driver started executing a maneuver. In this way, the label of each record changed its value to one (1), corresponding to a near-crash. New records continued to be saved with this label until the buttons of each device were pressed again. The new press of the button indicates the end of the near-crash driving maneuver, and further records were labeled as normal driving events (label equal to 0).

Each maneuver was executed a certain number of times. Table 2 shows the number of repetitions for each maneuver.

**Table 2.** Number of repetitions of each maneuver in the controlled tests.

| Type of Maneuver | Number of Repetitions |
| --- | --- |
| Sudden acceleration | 12 |
| Sudden braking | 9 |
| Aggressive right turn | 8 |
| Aggressive left turn | 7 |
| Aggressive line change to the right | 9 |
| Aggressive line change to the right | 11 |

The type of maneuvers or driving events presented in Table 2 were performed in fully controlled tests to train the ML algorithm. Table 2 shows the number of times each maneuver was performed in each of the controlled runs. These types of events commonly occur in trips made by vehicles and correspond to high-risk events.

2.5.2. Uncontrolled Validation Field Tests

Bearing in mind that the developed prototype seeks to help identify areas with a high probability of TAs, using near-crashes as a substitute measure for crashes, it is important to highlight that its operation and the obtained results are based on the frequency of occurrence of near-crashes in certain areas.

Achieving the above objective through the developed prototype implies the construction of a data set of vehicle kinematic variables. This data must be collected by performing trips on different key routes of the city in question, using a certain number of drivers, vehicles and defining other relevant factors for such tests.

The method selected to perform the trips through the city (field tests) followed the approach proposed by the ND studies. ND studies are characterized by providing insight into the behavior of the driver and his maneuvers throughout different trips made without any experimental control, maintaining his usual driving style. This is achieved with the collection of data associated with the driver, the vehicle and its environment.

The city selected as a case study was the city of Popayán, a medium-sized city in southwestern Colombia, with a population of approximately 300,000 inhabitants.

Including a large number of city trips and drivers, as well as a wide variety of vehicles in data collection generates an increase in the probability of recording a greater number of events of interest. However, a study with a large scope exceeded the limits of time and available resources (vehicles, drivers) of the work presented in this article. This led to a more selective inclusion of the city roads in which the data was collected.

Taking these limitations into account, and to maximize the probability of registering near-crashes during the field tests, it was necessary to perform a reconnaissance of the critical points and of the city roads that could present greater mobility conflicts. In this way, those roads and areas of the city where occurrence of mobility accidents was unlikely were initially ruled out.

For the selection of the critical points of the city, the information of the most recent mobility report of Popayán was reviewed. 15 critical points were identified.

The above information allowed a selection of the routes where a greater number of near-crashes could occur. These routes correspond to the paths where the uncontrolled field tests were performed. A total of 5 routes were defined, due to the difficulty of covering all the roads and points in a single route. These routes considered the most critical roads and mobility points. One of the defined routes is presented in Figure 6. Information about the 15 identified critical points and the 5 determined routes is available from the correspondence author upon request.

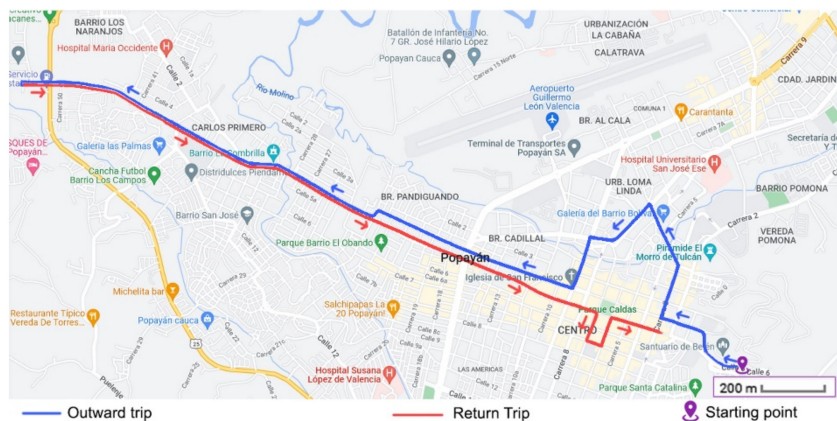

**Figure 6.** Route 1 map, distance: 11.2 km (map taken from Google Maps). The blue arrows indicate the direction of the vehicle's displacement on the outward trip of the route, the red arrows indicate the direction of the vehicle's displacement on the return trip.

The execution of these tests was performed by driving on the defined routes. For this, each route was covered by two drivers, each in their respective vehicle (Renault Logan model 2007 and KIA Picanto Ion model 2014). Both vehicles used during these tests met the two necessary requirements: belonging to the car category and being compatible with the ELM 327 OBD-II scanner.

For the execution of each trip, moments before starting each of them, drivers were asked to maintain their usual driving style. During the trips, the operators of the data collection devices accompanied the driver, avoiding as much as possible influencing their driving.

Data collection was performed simultaneously with each of the two considered devices: a Xiaomi Redmi Note 9S Smartphone with the mobile application installed and the developed hybrid device. Both devices collected the data using the same frequency used in the controlled training tests (20 Hz). These devices were in the intermediate area of the passenger floor, as shown in Figure 7.

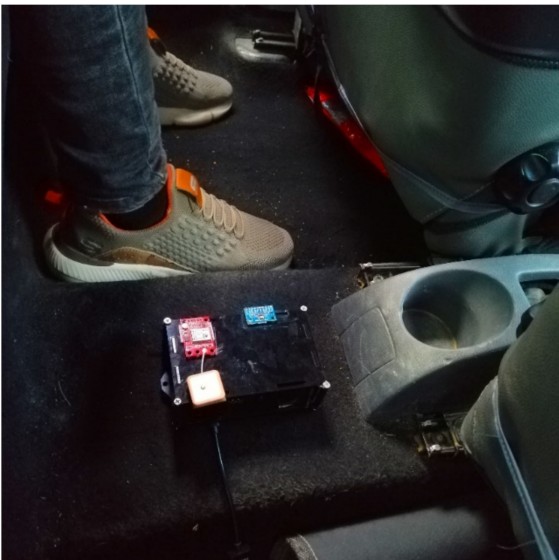

**Figure 7.** Location of devices inside the vehicle.

With the above conditions, data collection was performed for a total of 10 trips, and for each of these trips two data sets corresponding to each of the two data collection devices were created. This made it possible to compare their operation and performance under similar conditions. Table 3 briefly describes the execution of uncontrolled field tests.

**Table 3.** List of trips in uncontrolled field tests.

| Trajectory Id | Vehicle | Route | Date | Start Time | Duration | Number of Records |
|---|---|---|---|---|---|---|
| 1 | Kia Picanto Ion 2014 | 1 | 25 March 2022 | 4:16 p.m. | 55 min | 65,900 |
| 2 | Kia Picanto Ion 2014 | 2 | 30 March 2022 | 4:18 p.m. | 53 min | 64,040 |
| 3 | Kia Picanto Ion 2014 | 3 | 31 March 2022 | 11:23 a.m. | 43 min | 51,456 |
| 4 | Kia Picanto Ion 2014 | 4 | 1 April 2022 | 11:39 a.m. | 48 min | 58,206 |
| 5 | Kia Picanto Ion 2014 | 5 | 2 April 2022 | 4:08 p.m. | 51 min | 60,691 |
| 6 | Renault Logan 2007 | 1 | 7 April 2022 | 9:41 a.m. | 72 min | 85,884 |
| 7 | Renault Logan 2007 | 2 | 7 April 2022 | 4:17 p.m. | 66 min | 79,730 |
| 8 | Renault Logan 2007 | 3 | 6 April 2022 | 3:49 p.m. | 67 min | 80,560 |
| 9 | Renault Logan 2007 | 4 | 6 April 2022 | 9:33 a.m. | 48 min | 57,824 |
| 10 | Renault Logan 2007 | 5 | 1 April 2022 | 10:27 a.m. | 36 min | 43,909 |

## 3. Results

Next, the results obtained in the development of the ICRDS prototype, the results of the controlled tests (for the development of the ML model), and the results of the uncontrolled tests (for the validation of the ICRDS prototype) are presented, identifying the areas with the highest number of near-crashes.

### 3.1. ICRDS Prototype Development Results

The ICRDS prototype, as presented in Section 2.3, consists of two main modules: DCM and IIAM. The results obtained in the development of each of these modules are presented below.

#### 3.1.1. DCM Development Results

The OBE, which is the central component of the DCM module, had two instances in the prototype: a Smartphone and a hybrid device.

Each of the OBE instances included a different data set. Since the hybrid device also captures data from the vehicle's OBD-II interface, its associated data set includes an additional column, and the captured data differs from the data captured by the smartphone. The data sets collected for each of the two DCM module devices are described in detai in [43].

The results obtained in the development of each OBE instances are presented below:

- Smartphone:

The application developed for the Smartphone device has two interfaces for its operation, the first one is used for data collection and the second one presents the history of trips made and allows storing such trips in the central database.

The first interface is called "Data collection" and is presented in Figure 8a. This interface is made up of two different sections called: "Sensors" and "Travel information".

The "Sensors" section of the first interface has the function of showing the different values of the variables that are being collected, including their respective units of measurement.

The second section of the first interface, called "Travel information", was implemented to allow the DCM operator to adjust and select certain parameters. These parameters are the data capture frequency (expressed in Hz), the vehicle identifier and the identifier of the driven route.

Additionally, the "Travel information" section contains two buttons. One of them allows the DCM operator to control the start and end of the data collection process. The other one performs data labeling during the training phase of the ML algorithm. In this section a very important parameter for data collection is the frequency (in Hz). The frequency used in data collection of this device (Smartphone) and the hybrid device was 20 Hz. The selection of this value is justified in Section 2.5.1.

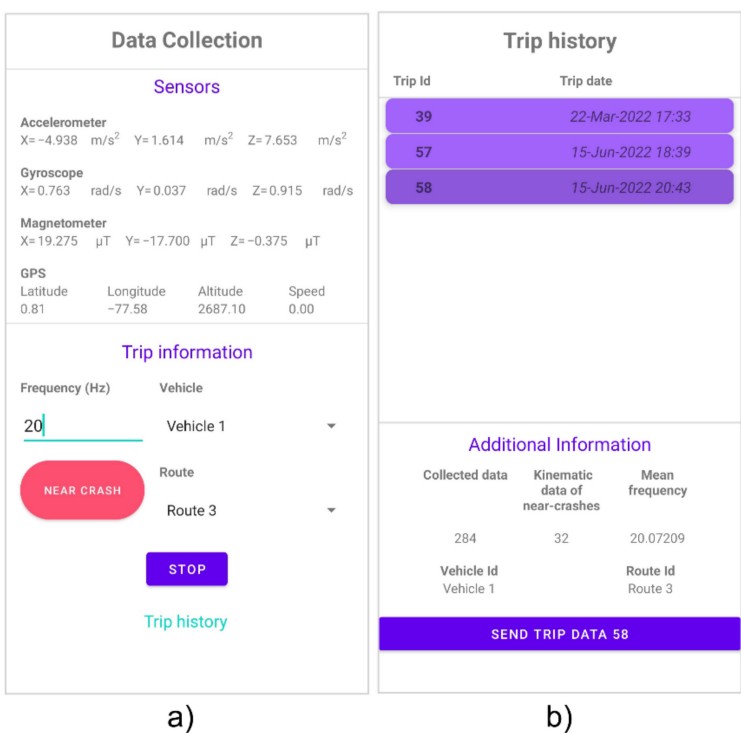

**Figure 8.** (**a**) "Data collection" interface in Smartphone application; (**b**) "Travel history" interface in Smartphone application.

While data collection is active, all variables are recorded in the SQLite database. The number of new records added to the database during one second is given by the data capture frequency.

The second interface is called "Travel history", which is presented in Figure 8b. This interface has two sections: list of routes taken and "Additional information" for each trip.

The list of trips made shows an ID, along with the data collection start date and time for each trip. The listed trips correspond to those on which a certain action has not been taken (delete or send to the central database).

The "Additional information" section of a trip includes the number of captured data records, the average frequency, the vehicle and route identifiers.

Finally, the second interface has a button called "Send travel data", which starts the process of extracting all the records belonging to the selected trip, to be sent to the central database.

- Hybrid device and web application:

The main component of this hybrid device is a Raspberry Pi 3B+ board, responsible for reading, pre-processing, managing and storing the collected data.

The Raspberry Pi board has a GPIO connector, which allows to integrate multiple types of sensors and HW modules in general. The hybrid device features the following hardware modules connected to the Raspberry Pi board:

1. An OBD-II ELM 327 scanner with Bluetooth connection, used to read the speed and the position of the accelerator pedal.
2. An IMU MPU 9250, used to read the kinematic values of the vehicle (acceleration, angular velocity and magnetic field intensity, in the "X", "Y", and "Z" axes). This IMU was connected to the Raspberry Pi using the I2C (Inter Integrated Circuit) serial communication port and protocol.
3. A GPS module Neo 6M, used to obtain the geospatial location of the hybrid device (latitude and longitude).

4. A pushbutton connected to a GPIO pin configured as an input. This button was used for data labeling during controlled data collection tests for training the ML algorithm.

Figure 9 presents a block diagram of the hybrid device.

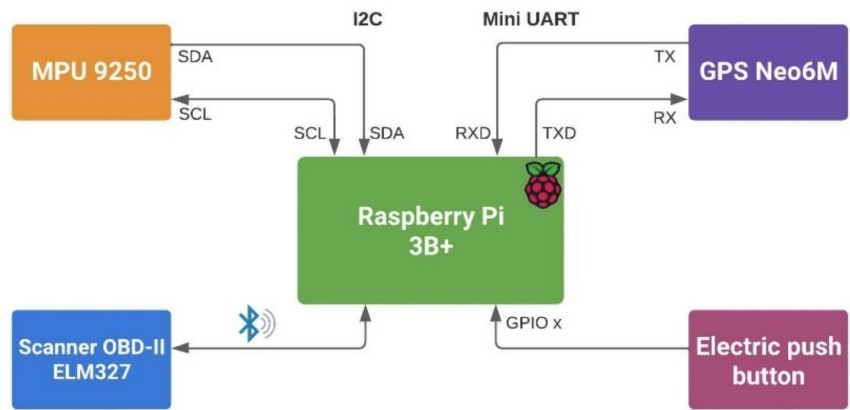

**Figure 9.** Hybrid device block diagram of ICRDS.

It was possible to provide a functionality similar to that described for the Smartphone application, by using a web interface. Two types of interfaces were also developed, an interface for "Data collection" and another for "Travel history".

3.1.2. IIAM Development Results

The IIAM module was divided into two main submodules: the "travel data management and analysis submodule" and the "results display submodule". The results obtained in each of the submodule are presented below.

- Travel data management and analysis submodule.

The components of this submodule allow the data administrator to read the central database (ADS-CD), to filter and pre-process the data (ADS-DFP) and to execute the ML model for classification of driving events (ADS-AAM).

This submodule has an interface that allows for the managing of the data collected on vehicle routes. In addition, there is a section that allows to view graphs to have a better understanding of the data. These graphs are a line chart, histogram, pie chart, and scatter matrix.

The last two functionalities available in this submodule are: downloading trip data in CSV format and checking for near-crashes. The latter is one of the most important as it allows a data manager to start the process of analyzing the data collected during a trip. During this process, such data is supplied to the ML model, which classifies it in search of near-crash events.

The result of the near-crash verification process is also stored in the central database. Such results include its location (latitude and longitude), date and time of occurrence, collection device, route, and the records that make up the near-crash. Figure 10 presents the main interface of the submodule.

- Results display submodule.

The main objective of this submodule is to display graphically the results of the entire collected data analysis process, mainly the near-crash checking process. This is achieved by using a heat map and a dot map in which the different near-crashes detected are located, allowing to observe the distribution of their occurrence over the different areas of the city under study. These maps were built using the plotly library in combination with OpenStreetMap [49].

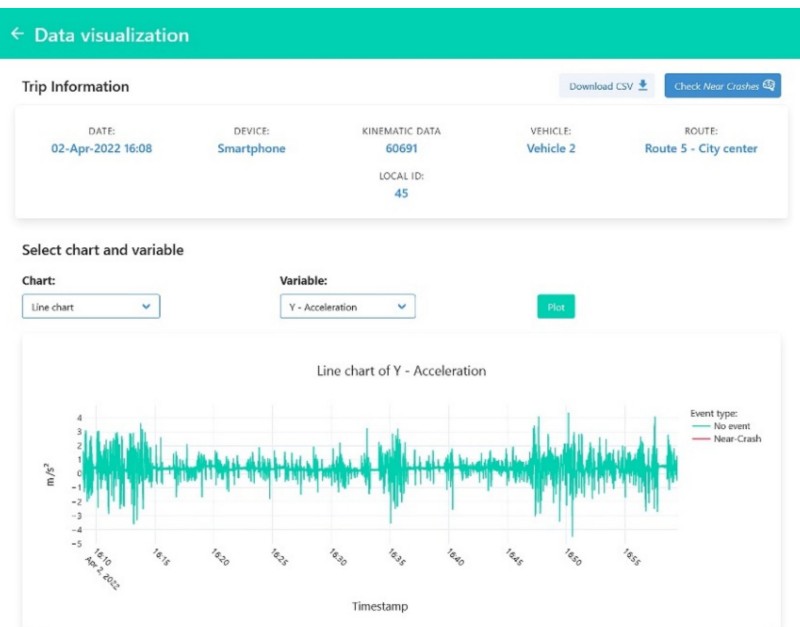

**Figure 10.** Information interface of the trip data collected in the ICRDS.

The web interface of this submodule allows the users to choose the map they want to observe, be it heat or dots. In addition, it provides the option to choose the data that is displayed on the map by selecting the device or a specific route. Examples of heat maps and dot maps are presented in Section 3.3.

The development of the ICRDS prototype was validated with help of the Scrum methodology, verifying compliance with all planned functionalities. Some adjustments were required in the hardware and web interfaces for full compliance.

### 3.2. Results of the Controlled Tests for the Development of the ML Model

The performance of the ML algorithm was obtained through the cross-validation technique and the optimization performed in phase 4 of the CRISP-DM methodology. The evaluation metrics used to quantify the performance of the algorithms were: precision, recall, F1-score and AUC.

The performance results of each algorithm provide the basis to select the algorithm to be used in the ICRDS system. These results were presented as a box plot, in which the distribution of the results could be observed after performing the cross-validation technique. This was done for the three selected algorithms (SMV, DT, and RF) and each of the two devices implemented (Smartphone and Hybrid device). In these box plots, results obtained by using a time window of 40 samples and another of 20 samples are shown in two different colors.

The evaluation of the three algorithms, in both devices, with different time windows, was performed using the first collected data set, through the planned controlled tests. Performance results for these tests (precision, recall, F1-score and AUC metrics) are presented in Appendix D.

For the Smartphone device, the results allow to state the following:

- The SVM algorithm reached a perfect precision in the maximum value of the box plot, but the RF algorithm had a better average in this metric.
- The SVM algorithm presented many variations in the distribution of the precision performance values, unlike the RF algorithm.
- The RF algorithm obtained the best performance in the results of the recall metric using a time window of 40 samples.
- The results of the recall and F1-score metrics were similar for the three algorithms, but in the case of F1-score, the SVM and DT presented several outliers.

- The performance achieved by the three algorithms in the results of the AUC metric had stable and high values (greater than 0.9).

  For the hybrid device, the results allow to state the following:

- The SVM algorithm reached a perfect precision in the maximum value of the box plot, but the RF algorithm had a better average for the case of a time window with 40 samples.
- The DT and RF algorithms performed well on the recall and F1-score metrics, while the SVM algorithm performed poorly.
- The SVM algorithm presented a large number of outliers in the results of the recall and "F1-score" metrics.
- The performance of the AUC metric was again the best for each of the algorithms, achieving values greater than 0.96 and with few outliers.

After the analysis of the data from the controlled tests and the evaluation of the algorithms, the RF algorithm was chosen for the development of the ML model due to its performance (the average of all the optimized metrics was 0.9801), little presence of outliers and uniform distribution in its results.

In addition, the 40-sample sliding window size performed better than the 20-sample window, so the 40-sample window was chosen.

It should be noted that in controlled tests identifying where the collision risk occurred is not useful. This type of test was performed with the exclusive objective of training the ML algorithm, so the events detected were intentionally performed in this test. For this reason, Section 3.2 does not present the specific identification of collision risks.

In contrast, the uncontrolled tests were performed to identify where the collision risks occurred using the ND method. Therefore, the detected events happened during the tests were not performed intentionally. For this reason, in Section 3.3, the specific identification of collision risks is presented.

### 3.3. Results of Uncontrolled Tests, Used to Identify Areas with the Highest Number of Near-Crashes and Areas with a High Probability of Accidents

The validation field tests allowed the construction of ten ND data sets for each of the devices used in data collection (Smartphone and hybrid device). Each data set corresponds to the trip made along a specific route in a certain vehicle. In total, the ten data sets of one device represent 539 min or approximately 9 h of ND; and a distance of 175.2 km.

By using the ML model developed, it was possible to analyze each data set seeking to identify near-crashes that occurred during the drives. Once this process was completed, it was possible to graphically observe the results through the heat maps and dot maps of the results visualization submodule. Figures 11–14 show the maps with the locations of the detected events.

A total of 349 near-crashes were detected in the data collected by both devices. A total of 81 of them were detected in the data collected by the Smartphone, and 268 near-crashes were detected from the hybrid device data.

To perform a more objective evaluation of the results, a comparison of these results was carried out with accident statistics for the years 2020, 2021 and the beginning of 2022, provided by Popayán's DMV.

The evaluation of the analysis results, performed on the data sets of each device, was based on the comparison of the number of near-crashes detected in the devices data with the number of accidents occurred in the same areas of the city.

The city zones that allowed the above comparison were defined by dividing the Popayán city area into a system of small grids. For this, the entire area of the city was initially defined within a square delimited by the intersection of a range of latitudes and a range of longitudes. The range of latitudes was established from 2.424 to 2.504; and the range of longitudes was established from −76.642 to −76.550.

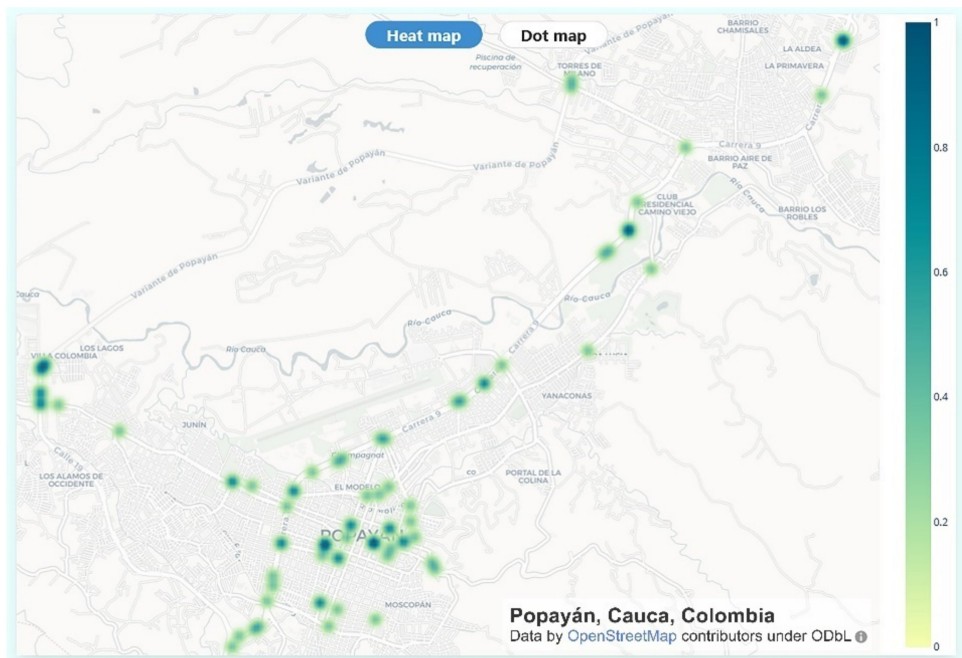

**Figure 11.** "Heat map" of the near-crashes detected in Smartphone data.

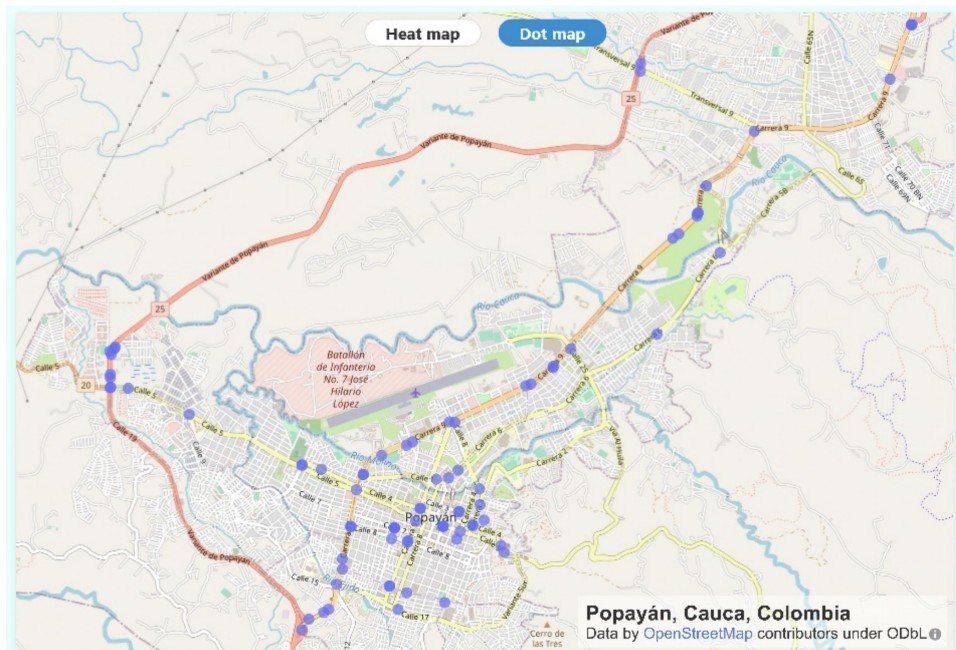

**Figure 12.** "Dot map" of the near-crashes detected in Smartphone data.

With a defined total area, it was possible to divide the city into several zones, corresponding to smaller grids, whose height and width extensions covered 0.004 degrees of latitude and longitude, respectively. In this way, all grid squares had the same size, al-lowing uniform distribution of the total area. Based on the distance measurement tool available on Google Maps, the side of each grid square had approximately 445 m.

For each of the areas identified on the city map, the number of near-crashes detected there was calculated. Following the explained grid system logic, 15 areas with the highest number of near-crashes, detected both from the Smartphone data and the hybrid device data, were identified. These zones are presented in Figures 15 and 16.

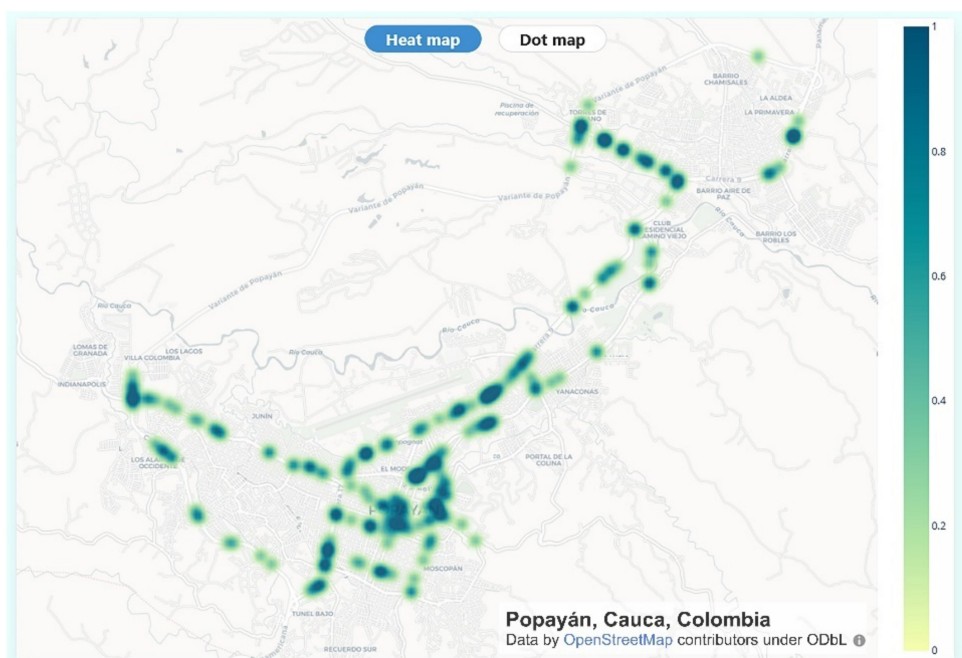

**Figure 13.** "Heat map" of the near-crashes detected in hybrid device data.

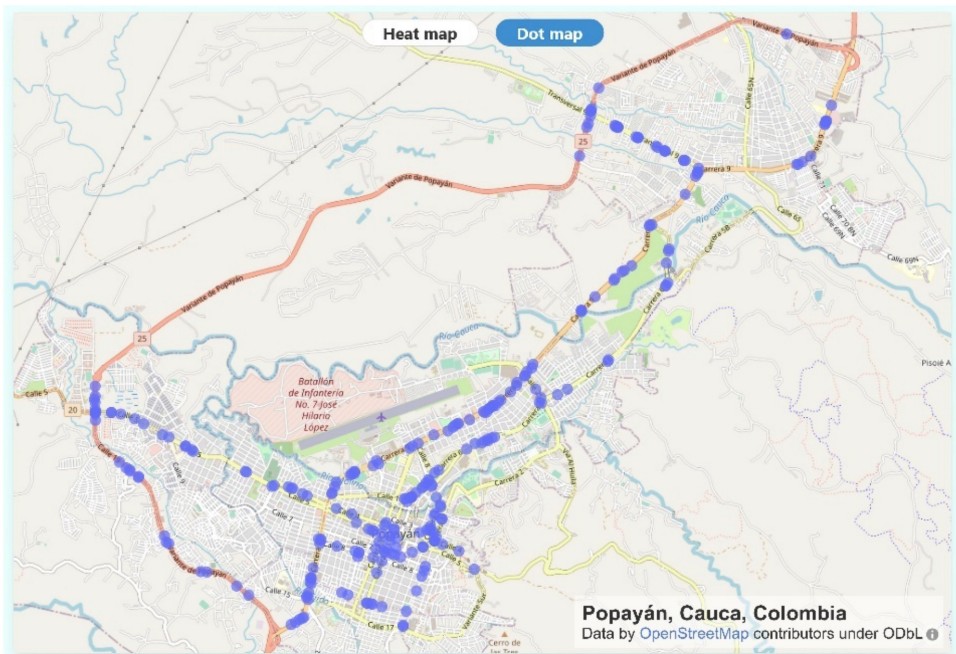

**Figure 14.** "Dot map" of the near-crashes detected in data hybrid device data.

Once the areas with the highest number of near-crashes were identified and plotted from the data from both devices, the areas with the highest number of TAs were plotted seeking to verify some quantitative relationship between the occurrence of near-crashes and the possible occurrence of a TA in a same area. Figure 17 shows the 20 areas with the highest number of TAs, according to the information provided by Popayán's DMV.

From reviewing the 15 areas with the highest occurrence of near-crashes identified from the Smartphone data and its ML models, it is possible to affirm that:

- A total of 12 of these areas coincided with one of the 20 DMV's reference zones.
- The top 4 of these areas continuously coincided with some reference zone. From the fifth zone, the coincidences occurred discontinuously or intermittently.

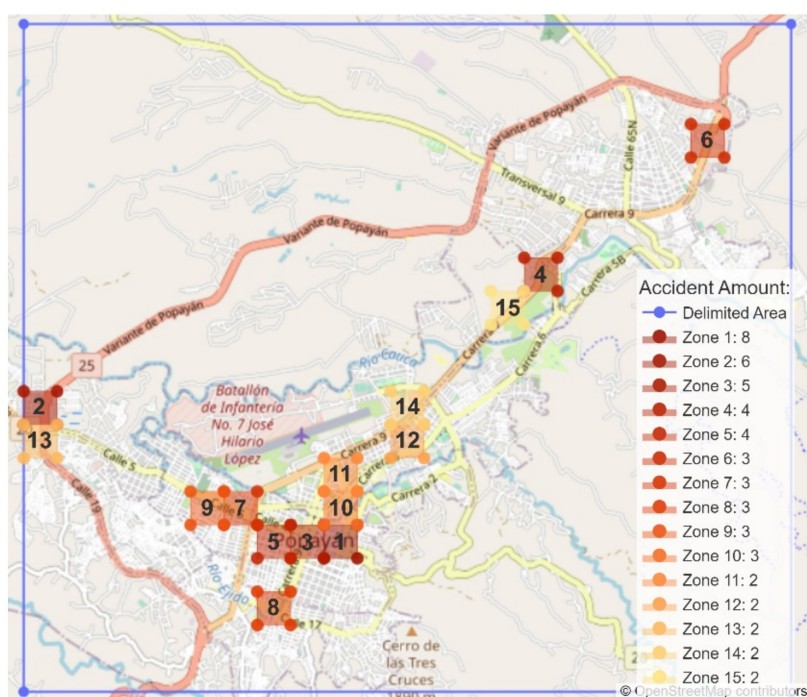

**Figure 15.** Map of areas with the highest number of near-crashes according to Smartphone data analysis.

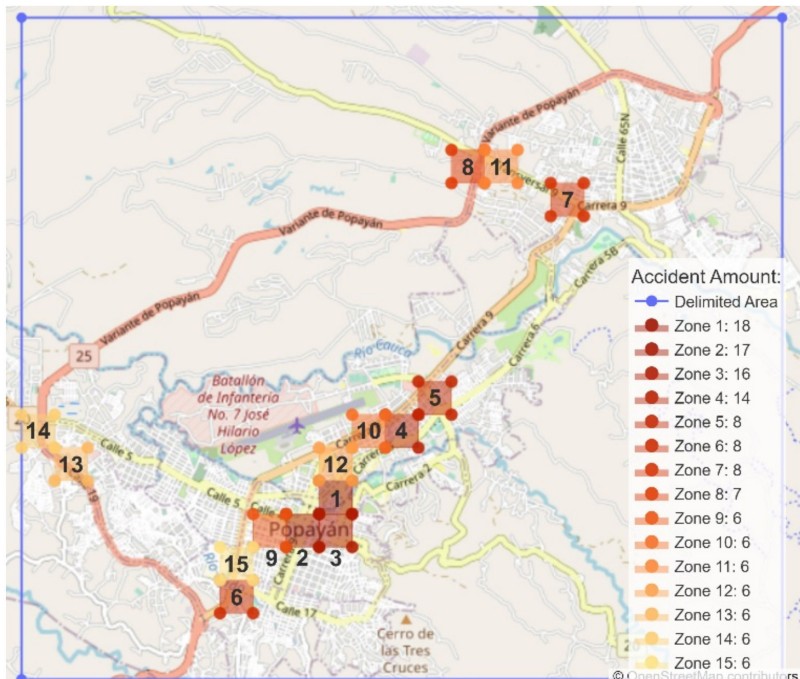

**Figure 16.** Map of areas with the highest number of near-crashes according to hybrid device data analysis.

Similarly, according to the hybrid device data and its ML models, it is possible to affirm that:

- A total of 8 of these areas coincided with one of the 20 DMV's reference zones.
- The top 5 of these areas continuously coincided with some reference zone. From the sixth zone, the coincidences occurred discontinuously or intermittently.

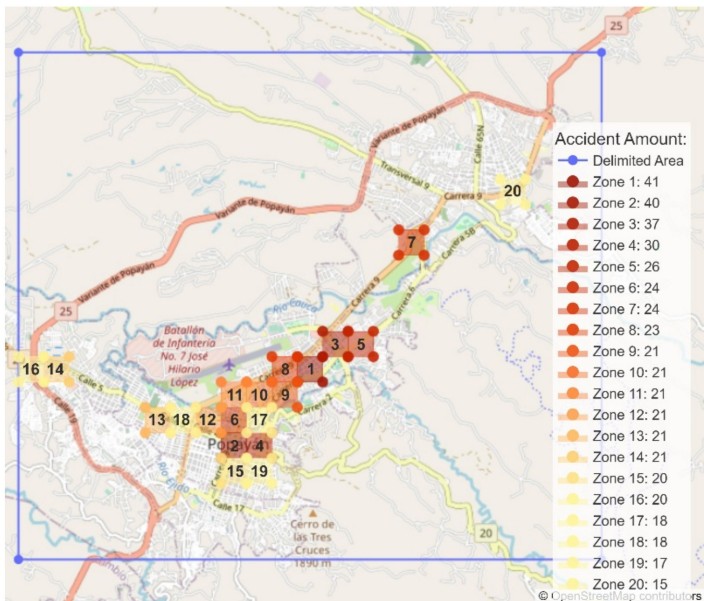

**Figure 17.** Map of areas with the highest number of traffic accidents according to data from Popayán's DMV.

According to the above, the distribution of near-crashes detected from the use of the Smartphone and its ML models had a greater similarity with the distribution of TAs occurred in recent years.

Moreover, it is possible to observe that the number of TAs that occurred in a reference area was not proportional to the number of near-crashes in its equivalent areas. This applied to both the Smartphone and the hybrid device.

## 4. Discussion

Next, two subsections are presented. The first discusses the results obtained (compared to those obtained in previous works), the constraints, and limitations of this work; and the second subsection discusses the contribution of the work to sustainability and safety in the transportation in a medium-sized city.

### 4.1. Results, Contraints and Limitation of This Work

The ICRDS prototype developed in this work previously identified the ideal components (data collection device, variables, and ML model) to apply it in the context of a medium-sized city in a developing country. However, it can be applied in any scenario in which it is desired to identify high-risk collision events, through vehicle trips.

Previous research works used contexts related to cities in developed countries, therefore, the evaluation of our target context for a work on transportation safety represents an important academic contribution [9].

By analyzing the performance of the developed ML models, it was possible to show that the results are very similar to those obtained by other related articles. This can be verified in the values obtained in the metrics performed in this work, compared with those obtained in previous works (accuracy, precision, recall, F1 score, and others). In addition, it was shown that the use of a relatively large time window to handle time series data increased the performance of the algorithms [14,18,19].

The analysis of the results obtained (once the ICRDS was implemented) was performed through a comparative evaluation of such results with accident data provided by Popayán's DMV. The results of this analysis show that there is a relationship between the areas with a high occurrence of near-crashes and the areas where TAs tend to occur more frequently. This was evidenced by observing that, regardless of the device with which the data was

collected, most of the areas with a high number of near-crashes agreed with some of the areas with the highest number of TAs.

Therefore, if it is considered the number of near-crashes detected from the data collected and the execution of the ML model, the hybrid device developed is more convenient than the Smartphone in detecting these events (with a difference of 187 near-crashes).

It is important to consider the following constraints and limitations of this work, to better evaluate the results obtained, and in future works seek to improve or increase some of them:

- The training of ML model was performed with data collected in controlled tests, trying to identify the six most frequent types of high-risk maneuvers. Although these tests had very good results, it would be convenient to retrain the ML algorithms with a larger data set, performing the controlled tests on the routes where the uncontrolled tests were performed. In this way, although they will continue to be controlled tests (where the event would be manually labeled as a "near-crash"), the environment of such tests will be more realistic, similar to that of uncontrolled tests. The proposed retraining is also suggested to use different filters and/or pre-processing techniques.

- Considering the experiments with the ND method that have been performed by previous works, it could be considered that the number of routes, vehicles, trips, and distance traveled in this work, were very low. However, it must be considered that the focus of the work was on medium-sized cities in developing countries, while previous work was performed in scenarios of developed countries. The low number of trips made in tests can be a reason for not identifying a direct relationship between the detected near-crashes and the TAs that have occurred in the city of Popayán in recent years. Considering this, it is recommended that for future work these parameters of the programmed tests be increased (both the controlled ones to train the ML model, and the uncontrolled ones, to validate the operation of the system prototype). Performing field tests with many more routes than those used in this work, employing more drivers, and making repetitions of the trips at different times, could help discover trends and relationships between the amounts of near-crashes and TAs for a certain area.

- The approach proposed and evaluated based on the ICRDS to identify areas with a high probability of TAs presents a clear advantage compared to traditional techniques for collecting statistical data on road accidents. The use of near-crashes and the ND technique allows data related to road safety to be obtained in shorter times and without the need for an incident to occur. This allows to analyze data and diagnose problems in a more agile way, and the results could be reflected in the implementation of TA prevention plans, modifications to mobility policies or improvements in road infrastructure.

- The proposed work identifies high-risk driving maneuvers, through the proposed sensors, using two types of devices (Smartphone and hybrid device). Although a rigorous process was performed for the selection of the variables to be measured and the sensors that would allow an adequate measurement, the possibility of including additional variables that allow the identification of the 6 maneuvers identified as high risk can be evaluated.

*4.2. Contribution to Sustainability and Safety in the Transportation of a City*

The ICRDS prototype developed allows the offline identification of high-risk driving events, indicating their exact location (longitude and latitude) in the city. This allows, through an analysis of the collected data, for the identification of high-risk areas in the city, where high-risk events have occurred on several occasions. The government entities in charge of analyzing this information can evaluate the causes of these high-risk events in the most critical areas of the city and determine if they can make decisions to improve this situation. For example, if near-crash events are repeatedly detected at a certain point where no adequate road signs are present, or the road condition is not good, or the posted speed limit is too high, corrective actions or policies can be taken to improve this situation.

The importance of traffic regulations and laws is illustrated by the reduction in deaths and injuries due to traffic accidents. According to WHO, Europe achieved a reduction of 13% in TAs between 2010 and 2016 [50].

By reducing the number of high-risk driving events (which can generate an TA), mainly the economic and social aspects of sustainability in city transportation are improved. TAs generate great social losses in a city, because these events are commonly associated with deaths and/or serious injuries. In addition, TAs commonly affect the flow of vehicular traffic in the city, for several hours, affecting the mobility of many citizens and their life quality. A reduction in the probability of injuries, deaths, and high congestion contributes positively to the social sustainability of transport in a city.

Regarding the economic aspect, TAs generate in most cases losses for the owners of the involved vehicles, and in some cases losses in the associated road infrastructure. Therefore, if the costs associated with TAs are reduced, it contributes positively to the economic sustainability of transport in a city.

The environmental aspect of the sustainability of city transport can be improved by reducing emissions. Driving maneuvers that occur in near-crashes such as sudden braking and acceleration generate a greater amount of vehicle emissions in a city [51]. If actions can be taken to reduce the probability of near-crashes in certain parts of the city, using as a basis the information generated by the developed ICRDS prototype, the amount of emissions could be reduced to a certain extent, improving the sustainability of transportation.

## 5. Conclusions and Future Work

The original hypothesis of the research work suggests that it is possible to determine the areas with a high probability of TAs, contributing to the sustainability and safety of transportation in a medium-sized city, using an ICRDS, which uses ML models for the analysis of data collected through naturalistic driving field tests. This hypothesis was validated through the development of the system prototype and the validation of the results.

The developed prototype is relevant for use in a medium-sized city in a developing country, considering the required budget, resources, and its performance. Its use would allow the identification of high-risk areas of collision in the city in an efficient way and the making of appropriate decisions to avoid these events.

The ICRDS prototype was designed and developed considering suitable alternatives for the most important parameters based on the literature review. This system was made up of a DCM and an IIAM.

During the development of the DCM, it was possible to verify that the development of the hybrid device turned out to be more complex than the development of the mobile application of the Smartphone, this is mainly due to the errors presented in the hardware of the device and the difficulty to successfully integrate all the components.

From the developed DCM, it was possible to collect ND data by executing validation field tests designed to identify areas with a high probability of TA in the city of Popayán. Subsequently, the IIAM made it possible to analyze all the collected data, revealing the locations of each one of the detected near-crashes.

Based on the development of the prototype and the presented results, the following proposals are made for future work:

- Scaling the DCM with functionalities such as a reorientation algorithm, applying architectures such as the publish-subscribe pattern to constantly send data to the cloud and in real time to the ICRDS web application. In this way, the prototype could adopt an approach based on IoT technologies and expand its possible applications.
- Scale the IIAM with functionalities such as data processing through tools in the cloud, execution of algorithms in the cloud and use of extract, load, transform (ELT) and extract, transform, load (ETL) tools for data management and transformation.
- Apply a new filter or optimize the Kalman filter used in the ICRDS, since the use of this filter considerably increased the data pre-processing time.

- Improve the ML model based on new data captured in ND field tests or retrain it using data from maneuvers executed by drivers with different driving styles.
- Address new ML techniques for the classification of near-crash events, these techniques could use unsupervised algorithms or neural networks.
- Execute validation field tests, by making more trips on new routes or on those designed for this work, using a greater number of drivers, to collect more ND data and perform a more complete evaluation of the system.
- Evaluate the usability of the system in entities such as DMVs or in entities related to the area of transport and road safety.

**Author Contributions:** Conceptualization, R.S.-C. and Á.P.d.l.C.; methodology, R.S.-C., Á.P.d.l.C. and J.M.M.M.; software, J.J.P.R. and S.F.Y.C.; validation, R.S.-C., J.J.P.R. and S.F.Y.C.; formal analysis, R.S.-C., J.J.P.R., S.F.Y.C., Á.P.d.l.C. and J.M.M.M.; investigation, R.S.-C., J.J.P.R. and S.F.Y.C.; resources, Á.P.d.l.C. and J.M.M.M.; writing—original draft preparation, R.S.-C., J.J.P.R. and S.F.Y.C.; writing—review and editing, Á.P.d.l.C. and J.M.M.M.; supervision, R.S.-C., Á.P.d.l.C. and J.M.M.M.; funding acquisition, Á.P.d.l.C. and J.M.M.M. All authors have read and agreed to the published version of the manuscript.

**Funding:** This research received no external funding.

**Informed Consent Statement:** Not applicable.

**Data Availability Statement:** The data presented in this study is openly available in Kaggle at https://doi.org/10.34740/KAGGLE/DS/2271186 (accessed on 1 August 2022) [43].

**Acknowledgments:** Authors wish to thank Universidad del Cauca (Telematics Department) and Universidad Icesi (ICT Department) for supporting this research.

**Conflicts of Interest:** The authors declare no conflict of interest.

**Appendix A**

In the architecture presented in Figure 3, two main modules are identified:

- Data collection module (DCM). This module groups the physical objects having functionalities related to sensing, storage and collection of vehicle data.
- Intelligent information analysis module (IIAM). This module groups the physical objects having functionalities related to analysis or management of previously collected vehicle data.

The physical objects presented in the architecture (Figure 3) are the following:

- Basic vehicle (BV). It represents the vehicle in operation, including its interfaces, platforms and electronic data on board the vehicle.
- Location and Time Data Source (LTDS). Device that provides the measurement of time and location.
- Vehicle On Board Equipment (Vehicle OBE). It provides the vehicle with sensory, processing, storage and communication functions based on the routes.
- Archived Data Administrator (ADA). It represents the operations performed by a human operator. In the proposed architecture, this object moves the data between the two modules.
- Archived Data System (ADS). Collect, archive, manage and distribute the data delivered through the ADA.
- Archive Data User System (ADUS). Object used to access, manipulate, analyze and process the archived data.
- Results Presentation System (RPS). Object used to visualize the information of the vehicular routes.

The functional objects presented in the architecture (Figure 3) are the following:

- Collected Data Storage (CDS). Allows to store the vehicular data collected on trips.

- Vehicle Kinematics Recording (VKR). Allows the collection of kinematic data through sensors on board the vehicle.
- Vehicle Location Determination (VLD). Allows determining the current location.
- Data Filtering and Preprocessing (DFP). Allows data filtering and pre-processing, to have reliable data for later analysis.
- Archived Analysis and Mining (AAM). Enables running advanced analysis, summary, and mining functions on large data sets.

The starting point of the system was the physical objects: BV and LTDS. These components provide the system with the current state of the vehicle's kinematics, current location and time. The Vehicle OBE provides some additional sensors, processing, storage, and communications. These functions would be performed while the vehicle is traveling.

The IIAM module has 3 physical objects. The first object (ADS) performs collection, archiving, management and distribution of data collected from different sources. This component can be seen as a centralized data warehouse.

The second component, the ADUS, includes systems such as databases, models, or analytical tools that a user will use to access the data. In this physical object the ML model capable of detecting near-crashes is established and trained. This component receives requests for advanced data processing, analyzes and delivers the results or products of any requested information processing.

The third component, the RPS, presents the results obtained from the information analysis. This component allows to visualize the results and any other information of interest in a suitable way.

Finally, it was necessary to include an actor within the architecture (the ADA). This user performs the data extraction from the DCM module and inputs it into the IIAM module. In addition, this user initializes the required processing tasks and monitors the obtained results.

**Appendix B**

The prototype presented in Figure 4 has two main modules: data collection and intelligent information analysis.

- Data Collection Module (DCM):

This module has the necessary hardware (HW) and software (SW) for the collection and storage of vehicle kinematic data. This process encompasses three main operations: capturing kinematic variables using the corresponding sensors, obtaining location and time data, and temporary storage of the collected data.

The DCM (bottom part of Figure 4) is mainly represented by the physical object called On Board Equipment (OBE). The OBE contains three functional objects: Collected Data Storage (OBE-CDS), Vehicle Kinematics Recording (OBE-VKR) and Vehicle Location Determination (OBE-VLD). The OBE is the physical object in charge of obtaining the kinematic data of the host vehicle, which are captured from the use of an accelerometer, gyroscope, magnetometer and OBD-II. In addition, it obtains location and time data, obtained from the GPS and internal clock of the device.

The analysis of alternatives evaluated for the ICDR system (described in Section 2.1) determined that a Raspberry Pi 3 board and a Smartphone were proposed as the devices to be used for the OBE. Each of these devices corresponds to a different instance of the OBE.

It was considered convenient to use two devices at the same time to measure the variables, to allow data and performance comparisons.

The use of a database engine was considered to provide a standardized and common temporary storage mechanism for the collected data for both instances of the OBE. This database engine was the basis for the development of the OBE-CDS functional object.

- Intelligent information analysis module (IIAM):

This module contains the required components for processing of collected information and data cleaning, ML algorithms and presentation of results (top part of Figure 4). The

module is made up of two physical objects: ADS, which has the logic, algorithms and software necessary for data processing; and the RPS, in which the areas with the highest probability of occurrence of a TA are identified, according to the near-crashes detected.

The ADS object processes the information obtained by the OBE and learns from it. For this, it uses a central data set that collects the measured data of the vehicle's trips. This data set is pre-processed and post-processed using appropriate Python libraries. Furthermore, this physical object includes an ML model that detects the occurrence of a near-crash.

The RPS physical object presents the results obtained by the OBE and the ADS. It consists of a web application (ICRDS Web), where the information resulting from the implementation of the field tests and the output of the ML model is intuitively displayed. The web application was implemented with technologies such as HTML, CSS, JS and Flask. The use of these technologies allows the presentation of a heat map and a dot map of the city under study, indicating the locations where the different near-crashes occurred.

The prototype includes a physical object (actor) outside of the previous modules, this is the ADA. The ADA moves the data from the DCM module to the IIAM. The process has two steps: (a) Extraction of the temporary data of a trip, which is stored in the Raspberry Pi or the Smartphone. (b) Store all trip records with their respective data in a central database.

The database, a Realtime Database (non-relational Firebase database [52]) is deployed in the cloud. Firebase is the Backend as a Service (BaaS) offered by Google, which has become a unified platform that facilitates the development of Android, IoS and web applications [53]. This database allowed for easy transformation to a Comma Separated Values (CSV) format, making exploration in the IIAM module easier. The CSV format is used for its simplicity and its massive use in ML libraries in languages such as Python [54].

**Appendix C**

Six functional objects were implemented in the prototype, distributed in the physical objects previously explained. These functional objects are described below.

- OBE—Collected Data Storage (OBE-CDS):

It corresponds to an SQLite database installed both on the Smartphone and on the Raspberry Pi board. This database has the function of storing the data coming from the sensors, the vehicle's OBD-II system, the device's internal clock and the GPS.

SQLite is a library written in C language, which implements an SQL database engine with many advantages: it is small, fast and reliable [55].

In addition, this functional object includes a software in charge of receiving the kinematic, location and time data provided by their respective sources, and storing them into the database.

- OBE—Vehicle Kinematics Recording (OBE-VKR):

This functional object corresponds to a software in charge of reading the vehicle kinematics variables measured by the sensors (accelerometer, gyroscope and magnetometer) and the OBD-II system in the case of the Raspberry. Subsequently, it supplies this data to the OBE-CDS, to be recorded in the database. This software module is part of a mobile application (in the case of the Smartphone) and a program written in Python (in the case of the Raspberry Pi 3 board).

- OBE—Vehicle Location Determination (OBE-VLD):

Software responsible for supplying to OBE-CDS with a timestamp corresponding to each vehicle data record and its current location. These data are obtained from the internal clock and GPS included in the OBE. As with the OBE-VKR, this is part of a mobile application and a program written in Python according to the OBE instance.

- ADS Central Data set (ADS-CD):

It corresponds to the Firebase Realtime Database instance and the set of files in JSON format, which contain all the data collected from the different vehicles, which have been previously extracted from their respective OBE-CDS.

- ADS Data Filtering and Preprocessing (ADS-DFP):

This functional object allows the cleaning and pre-processing of the data stored in the ADS-CD. This is done using Python's Pandas library, that allows reordering, operating, cleaning lost data, among other actions. This library works together with other libraries that can be used for data preparation.

- ADS Archived Analysis and Mining (ADS-AAM):

This functional object analyzes and processes the data prepared by the ADS-DFP and learns from it. This functional object contains the implementation of the ML model, made with the ScikitLearn library [56]. The output of this object is the binary classification of an event as a near-crash or a no near-crash.

For planning and monitoring of the development of both modules of the prototype (DCM and IIAM), the agile development methodology SCRUM [57] was used. In the initial phase of this methodology, the prototype was identified as the principal part of the business case to be presented to the stakeholders, as well as for the construction of the project vision. Each feature was initially described as an epic, and later as a user story. The development of the entire project was addressed in three sprints.

For the development of the data analysis process and ML model, which is responsible for analyzing the collected data sets and determining if they contain near-crashes, the Cross Industry Standard Process for Data Mining (CRISP-DM) methodology was used. CRISP-DM is a methodology for the development of data mining processes [58], which provides a life cycle model for data analysis projects. CRISP-DM has 6 phases: business understanding, data understanding, data preparation, modeling, evaluation, and deployment.

**Appendix D**

The performance of the algorithms using input data from the Smartphone device, during the performed controlled tests, is presented below. Figures A1–A4 show precision, recall, F1-score and AUC metrics, respectively.

Figures A5–A8 show the performance of the algorithms with data collected from the Hybrid device. Precision, recall, F1-score and AUC metrics are shown.

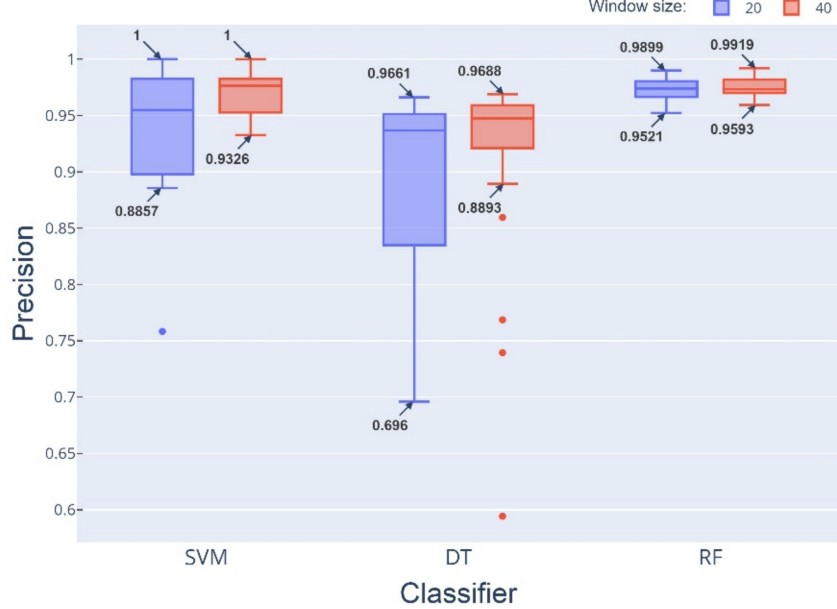

**Figure A1.** Box plot, performance of the "precision" metric for the Smartphone device data.

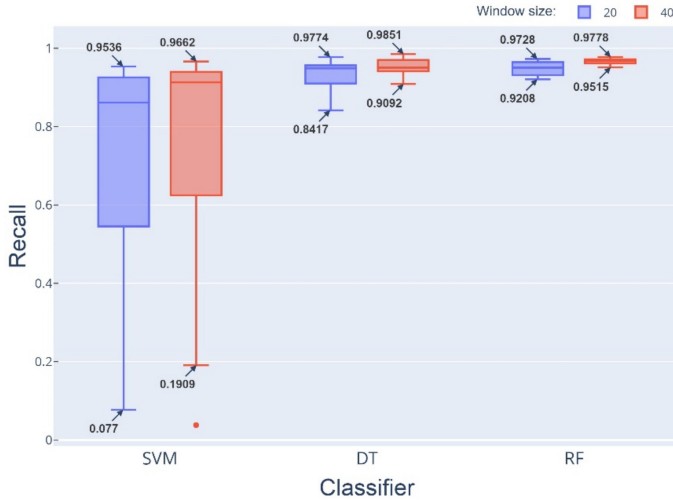

**Figure A2.** Box plot, performance of the "recall" metric for the Smartphone device data.

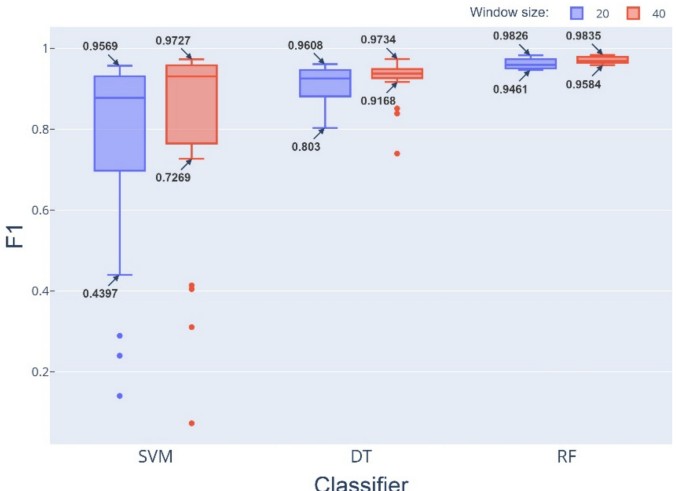

**Figure A3.** Box plot, performance of the "F1-score" metric for the Smartphone device data.

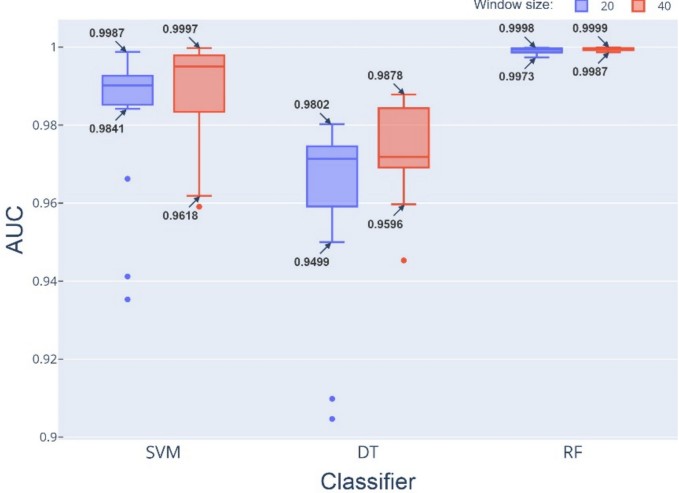

**Figure A4.** Box plot, performance of the "AUC" metric for the Smartphone device data.

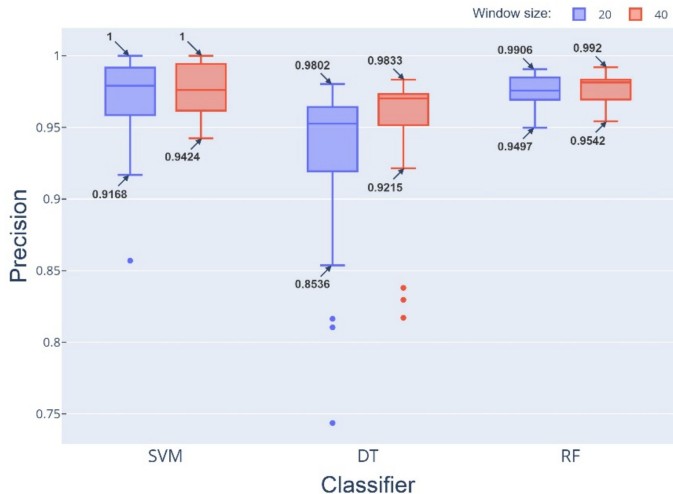

**Figure A5.** Box plot, performance of the "precision" metric for the hybrid device data.

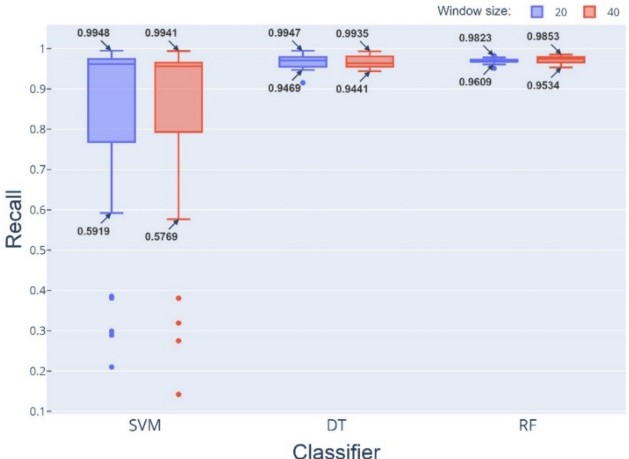

**Figure A6.** Box plot, performance of the "recall" metric for the hybrid device data.

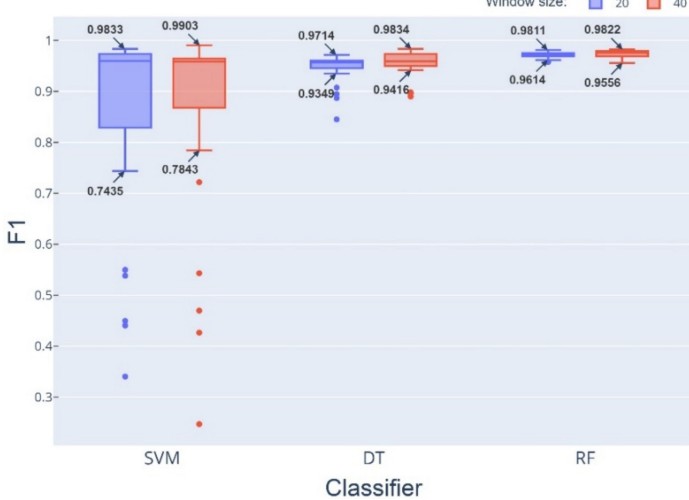

**Figure A7.** Box plot, performance of the "F1-score" metric for the hybrid device data.

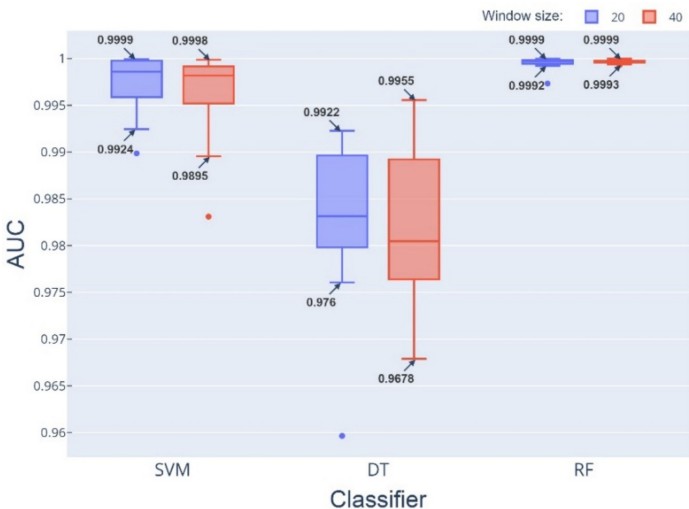

**Figure A8.** Box plot, performance of the "AUC" metric for the hybrid device data.

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
