# Peer review of "Design, Development and Validation of an Intelligent Collision Risk Detection System to Improve Transportation Safety: The Case of the City of Popayán, Colombia"

_sustainability, doi:10.3390/su141610087_

Round 1
Reviewer 1 Report
This paper presents collision risk detection system to improve transportation safety using naturalistic driving data and machine learning techniques. Although the title seems very interesting, the contents and contributions of the work are a bit ambiguous. I could not find information for the following points:
What kinds of collision risks are considered? For rear-end collisions, car-to-car driving behavior should be the main factor instead of considering entire city traffic.
Where should the proposed system be implemented, on each car or on some monitoring cells?
None of the novelty claimed in the five itemized statements can be considered contributions since they only provide abstract information.
What kind of data are generated by ND? Do these data include car following gaps, merging, and turning data of individual vehicles? What context is used, i.e., the number of lanes, curvature, congestion level? Provide some specific examples of collision risks and associated data. Since the data is generated manually, how the risk of collisions is defined. Unless the data represent realistic scenarios, the developed system cannot be trustworthy.
Why is it necessary to address medium-sized cities? Does the system work for small or big cities too? The same question is for developing countries. Motivation to choose those specific targets should be described with justifications, e.g., providing accident data. It is unclear how the considered ND data reflects the considered scenarios, as no example is given.
Various types of data, including device, acceleration, GPS, and trip history, are described. How are they fed into an ML model? The architecture of ML model, with input and output, is missing.
The phenomena provided in Table 1 can be found in any type of city or country to some extent. How do they vary for city sizes or countries?
The results should include some specific identification of collision risks, instead of only showing the algorithm performance.
Despite being an interesting topic, overall, the contents are very ambiguous without specific contributions. The proposed approach is not clearly described, and reproducibility is impossible.
Reviewer 2 Report
Page 1, Title of the paper: My suggestion is to change the existing title “Intelligent collision risk detection system to improve transportation safety in medium-sized cities of developing countries: design, development and validation” to “Design, development and validation of an intelligent collision risk detection system to improve transportation safety: the case of the city of Popayán, Colombia”. The proposed title better reflects the content of the paper.
Page 1, Keywords: My suggestion is to include “Machine Learning” in the Keywords.
Section 1. Introduction: Please include some more details concerning the 1st paragraph (lines 34-37). For example, the reference year for the presented data etc.
Section 1. Introduction: Please justify within your manuscript why “transport infrastructure/road infrastructure” was not included among “the reasons for high burden of road traffic injuries in developing countries”.
My suggestion is better justifying in Section 1. Introduction the reason(s) why you have decided to focus your research to developing countries.
Page 2, the last 3 paragraphs, lines 83-98: Please add much more text concerning the 8 references [8-15], the 6 references [16-21], the 5 references [22-26] and the 8 references [27-34]. These topics are very interesting for the reader. I strongly believe that the Introduction will be substantially improved with the additional text.
Please specify within the manuscript (at the end of Section 1. Introduction) the research questions concerning your work.
Page 3, Section 2. Materials and Methods: My suggestion is to include a Data Flow Chart (DFC) describing all your methodological steps, so that the reader can have a clear overview of your work from the early beginning of your paper. You may use parts of Figures 1, 2 and 3 in the proposed DFC.
My suggestion is to include some photographs concerning the tests with the Nissan March model 2016, the Renault Logan 629 model 2007 and the KIA Picanto Ion model 2014, for the benefit of the reader.
My suggestion is to include the source of the geographical background (map) for Figures 16, 17, 18, and 19.
My suggestion is to have 2 separated Sections, one for the “Discussion” and one for the “Conclusions”. In the Section of “Discussion” please try to discuss your findings vs the respective findings in the literature review. In addition, please try to discuss further the constraints and limitations of your work (page 27, lines 966-972).
You mentioned on page 27, lines 930-931 that “…contributing to the sustainability and safety 930 in transportation of a medium-sized city…”. My suggestion is to write a large subsection concerning this statement. Since this paper was submitted to the prestigious Journal “Sustainability” it is very important to analytically explain why your work has an impact to sustainability, how this can be achieved, who will benefit from your work (in economic, environmental, and social terms) etc.
Reviewer 3 Report
This paper mainly introduces a tool or platform that users/agencies can be used to detect the probability of near-miss crashes using Naturalistic Driving data. This topic is vital for safety research since near-miss crashes are hard to be detected and recorded. Such a detection system would make it easy and effective for both data collection and analyses. Overall, this paper is well written and easy to follow. Unfortunately, I don't feel this paper is suitable for publication as a journal paper. Instead, it reads more like the instruction of the software package or application.
Reviewer 4 Report
1. The authors were recommended to clearly define the terms, such as low income city, medium-sized cities, etc.
2. Please provide more evidence about the statement “The burden is disproportionately borne by pedestrians, cyclists and motorcyclists, in particular those living in developing countries. ”
3. The authors were recommended to revise the last two academic contribution statements. From my perspective, the last two contributions fail to describe your own academic contributions.
4. The business understanding section was difficult to follow. Please provide more explanations.
5. How did the authors collect variables to represent distinct features in your study?
6. The following studies were recommended to be properly cited: [1] Sensing Data Supported Traffic Flow Prediction via Denoising Schemes and ANN: A Comparison. IEEE Sensors Journal, 2020. 20(23): p. 14317-14328. [2] A Data-Driven Method Towards Minimizing Collision Severity for Highly Automated Vehicles, IEEE Transactions on Intelligent Vehicles, vol. 6, no. 4, pp. 723-735.
Round 2
Reviewer 1 Report
The authors have revised the manuscript based on reviewers’ comments but not in a very organized way. It increases the length with the loss of links of description. The entire Introduction needs to be streamlined in a concise way. The paper is unnecessarily lengthy, and it is very hard to extract the part or information related to the main development of this work. The basic description can be removed or given in Appendix.
The three academic contributions claimed in the Introduction do not reflect what is stated in the title of the paper or in the Abstract. Though this reviewer still thinks these are not technical contributions, the title and the Abstract should also reflect the contribution of the paper consistently. I still feel a major revision is necessary to improve the readability of the paper.
Following are the additional comments:
Authors claimed in the Abstract: “an intelligent collision risk detection system using ND and ML to improve sustainability and safety in the transportation of medium-sized cities”. And in the response letter: “identify if the collected data in a certain time interval contains high-risk collision events”.
The targeted type of collision identification and scope should be clearly stated in the Abstract. If the risk of collision is identified after some interval (say 1 min or only after collecting 40 data samples) then how do these analyses link to the sustainability or safety improvement? It is not an online risk identification. Collision risks are often linked to randomness in driver behavior.
The list of the risky event shown in Figure 2 should be explained with specific criteria as they are conducted artificially, e.g., what is the threshold of sudden acceleration and sudden braking.
What is the frequency of collection 18 x 36,353 raw data points? Are they collected at each fraction of a second? Figure 5 should include X-axis labels. How are the near crash data identified to mark it 1 or 0? In my view, often, a vehicle may make sudden braking or turn depending on many factors. Characterizing all such cases as near collisions is very misleading. It is very necessary to justify using the examples of how a near collision case is marked to train the ML model.
Are they fed to the ML framework with 18 x 40 (20) inputs at once, in a sliding mode? Is the identification for each set of input or collectively for entire data? The training process should be summarized very clearly with full information in the respective section.
Reviewer 2 Report
I would like to express my deepest thanks to the authors because they have carefully addressed my comments.
Author Response
Thank you very much for the comments sent. They allowed us to considerably improve our article.Reviewer 4 Report
My comments have been addressed.
Author Response
Thank you very much for the comments sent. They allowed us to considerably improve our article.
Round 3
Reviewer 1 Report
The authors have made efforts to revise the manuscript according to the reviewer's comments.